# A stable quasi-solid electrolyte improves the safe operation of highly efficient lithium-metal pouch cells in harsh environments

Zhi Chang[1], Huijun Yang[1,2], Xingyu Zhu[1,2], Ping He [3] & Haoshen Zhou [1,2,3 ✉]

Nanoconfined/sub-nanoconfined solvent molecules tend to undergo dramatic changes in their properties and behaviours. In this work, we find that unlike typical bulk liquid electrolytes, electrolytes confined in a sub-nanoscale environment (inside channels of a 6.5 Å metal-organic framework, defined as a quasi-solid electrolyte) exhibits unusual properties and behaviours: higher boiling points, highly aggregated configurations, decent lithium-ion conductivities, extended electrochemical voltage windows (approximately 5.4 volts versus Li/Li$^+$) and nonflammability at high temperatures. We incorporate this interesting electrolyte into lithium-metal batteries (LMBs) and find that LMBs cycled in the quasi-solid electrolyte demonstrate an electrolyte interphase-free (CEI-free) cathode and dendrite-free Li-metal surface. Moreover, high-voltage LiNi$_{0.8}$Co$_{0.1}$Mn$_{0.1}$O$_2$//Li (NCM-811//Li with a high NCM-811 mass loading of 20 mg cm$^{-2}$) pouch cells assemble with the quasi-solid electrolyte deliver highly stable electrochemical performances even at a high working temperature of 90 °C (171 mAh g$^{-1}$ after 300 cycles, 89% capacity retention; 164 mAh g$^{-1}$ after 100 cycles even after being damaged). This strategy for fabricating nonflammable and ultrastable quasi-solid electrolytes is promising for the development of safe and high-energy-density LIBs/LMBs for powering electronic devices under various practical working conditions.

---

[1] Energy Technology Research Institute, National Institute of Advanced Industrial Science and Technology (AIST), 1-1-1, Umezono, Tsukuba 305-8568, Japan. [2] Graduate School of System and Information Engineering, University of Tsukuba, 1-1-1, Tennoudai, Tsukuba 305-8573, Japan. [3] Center of Energy Storage Materials & Technology, College of Engineering and Applied Sciences, Jiangsu Key Laboratory of Artificial Functional Materials, National Laboratory of Solid State Microstructures, and Collaborative Innovation Center of Advanced Microstructures, Nanjing University, Nanjing 210093, P. R. China. ✉email: hszhou@nju.edu.cn

Since their invention, rechargeable batteries have experienced great popularity in recent decades[1,2]. Batteries, especially lithium-ion batteries (LIBs)[3,4], are widely used to power various electronic devices ranging from portable electric devices such as wristwatches and smartphones to laptops and even electric vehicles (EVs). The rapid development and ever-increasing use of electronic equipment puts great demands on not only the densities and cycling lives of LIBs but also LIB production and shipment. According to a previous report, both the global LIB market and LIB shipments have gradually increased over the past three years and are expected to steadily climb in the upcoming five years[5]. Specifically, the global LIB market and shipments are predicted to be worth 108.9 billion dollars and 439.3 GWh in 2025[5]. In addition to the prospective prosperity of the LIB market and production expectations, improving the energy density of batteries is another effective way to meet the high demand for energy-consuming electronic devices.

Replacing the graphite anode (372 mAh g$^{-1}$ theoretical specific capacity for LIBs) with lithium metal (3860 mAh g$^{-1}$ theoretical specific capacity, an order of magnitude higher than graphite) can effectively improve the energy density of batteries (such as lithium-metal batteries, LMBs)[6]. For this reason, LMBs have attracted a high degree of research interest over the past several years[7,8]. Among LMB-related works, researchers have mainly focused their attention on preparing various functional electrolytes to improve the capacity and working time of LMBs[9–18]. Generally, LMBs assembled with typical liquid organic electrolytes tend to experience several long-lasting inherent and tricky problems that hinder their further development[19]. LMBs, especially high-voltage LMBs assembled with typical liquid electrolytes, tend to suffer from high levels of electrolyte degradation induced by the high reactivity between liquid electrolytes and charged transition metal oxide surfaces and reactive lithium metal (Fig. 1a)[10,20]. Specifically, at the

cathode, during electrochemical cycling processes, the high-voltage cathodes can be easily attacked by HF, which is dominantly produced by the hydrolysis of electrolytes, consequently resulting in more severe transition metal dissolution (TM loss)[21]. Additionally, an undesirable cathode electrolyte interphase (CEI) and apparent cathode reconstruction phenomenon also commonly occur during electrochemical charging/discharging processes[7]. At the lithium-metal anode, the constant electrolyte-lithium interaction combined with the transition metal ions shuttled from the cathode would aggravate lithium-metal anode degradation and lead to an unfavorable solid electrolyte interphase (SEI) and uncontrollable growth of dendritic lithium[12]. These detrimental issues that occur in LMBs assembled with a liquid electrolyte accelerate the failure of LMBs. Even worse, various gaseous products produced during the charge/discharge processes and the potential short circuit after the dendritic lithium pierces the separator can create serious safety hazards such as battery burning or even drastic battery explosions, especially when batteries are working at high temperatures[22]. Therefore, to further promote the development of LMBs, the aforementioned issues need to be reasonably addressed.

A gradual reduction in the amount of flammable liquid organic electrolyte added to the LMBs and a transformation of the electrolyte system into a solid state system is likely to solve these difficult challenges[23–26]. Solid-state electrolytes possess several apparent advantages over typical liquid electrolytes: they are much more thermally stable, they have a wider electrochemical stability window, the possibility of dendritic lithium formation is much lower, and nearly no electrolyte decomposition and/or volatilization occurs[27,28]. These properties enable solid-state electrolytes to significantly improve the safety of LMBs after assembly. However, they also have obvious inherent defects; for example, they have a much lower ion conductivity and inferior interfacial properties when coupled with electrodes[28,29]. In addition, the large-scale production of solid-state electrolytes

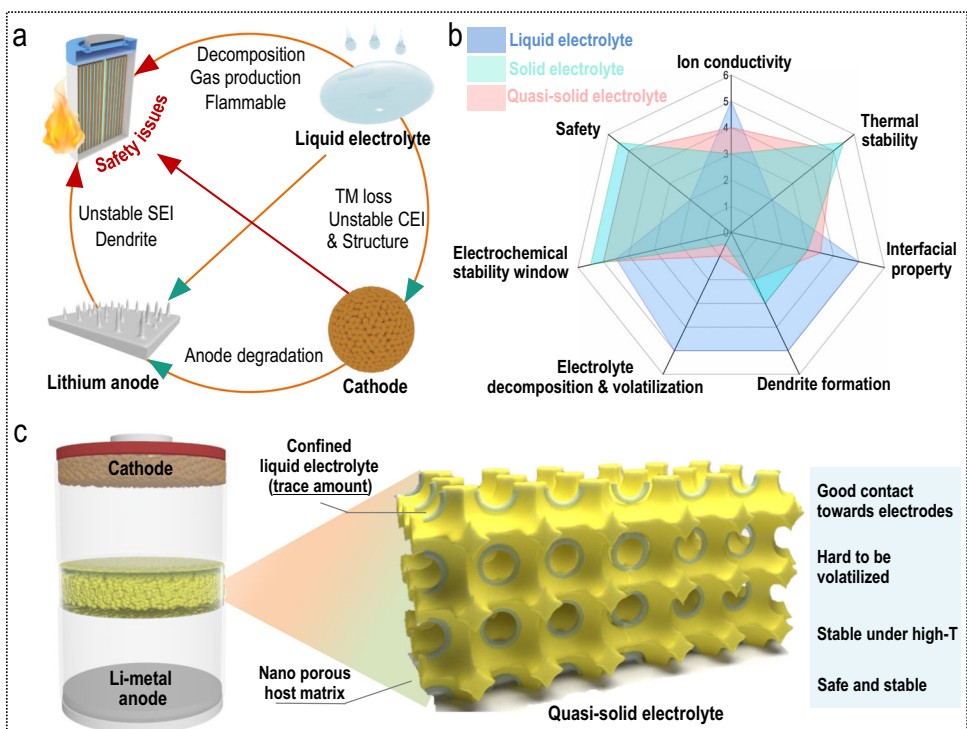

**Fig. 1 The importance of using a nonliquid electrolyte for safe lithium-metal batteries. a** Safety hazards induced by using traditional liquid electrolytes. **b** Radar chart of the advantages of quasi-solid electrolytes (compared with typical liquid electrolyte and solid-state electrolyte). **c** Lithium-metal battery assembled with the proposed quasi-solid electrolyte.

remains difficult, and their brittleness further limits their application[28]. Because they are in an intermediate state between liquid electrolytes and solid electrolytes, quasi-solid electrolytes have the advantages of liquid electrolytes and solid electrolytes while avoiding the shortcomings of both sides (Fig. 1b)[24,27]. Quasi-solid electrolytes can not only provide mechanical stiffness to block dendrites but also create a much safer (nonflammable) operation environment than typical liquid electrolytes. In addition, quasi-solid electrolytes also possess higher ion conductivity and superior interfacial properties than solid-state electrolytes. Confining small amounts of liquid electrolyte inside a host matrix that possesses a nanoporous (sub-nanoporous) structure, as shown in Fig. 1c, is a promising method of preparing a quasi-solid electrolyte that can meet the above requirements illustrated in Fig. 1b: good contact with electrodes, low volatility, stability and safe operation at high working temperatures.

Here, in this work, we have prepared a safe quasi-solid electrolyte that can enable high-voltage lithium-metal pouch cells to work normally and stably in a harsh working environment of a high working temperature (90 °C) even after sustaining damage (bent and cut). Confined inside the sub-nanochannels of the flexible and porous metal-organic framework (6.5 Å MOF), the quasi-solid electrolyte (total mass: ~3.5 mg cm$^{-2}$) contained merely trace amounts of liquid electrolyte (<0.23 μL cm$^{-2}$, equal to 0.3 mg cm$^{-2}$), which were apparently much lower than the amounts of the typical liquid electrolyte (total mass including separator: ~34.0 mg cm$^{-2}$) used for LMB assembly (~25 μL cm$^{-2}$, equal to 32.6 mg cm$^{-2}$). The quasi-solid electrolyte also demonstrated its advantage in weight when compared with a typical pouch cell assembled with lean electrolyte (3.5 mg cm$^{-2}$ vs. 4.5 mg cm$^{-2}$). Despite decent lithium-ion conductivity, the prepared quasi-solid electrolyte also demonstrated a wide electrochemical stability window (approximately 5.4 volts versus Li/Li$^+$). As a result, high-voltage LiNi$_{0.8}$Co$_{0.1}$Mn$_{0.1}$O$_2$//Li (NCM-811//Li with a high NCM-811 mass loading of 20 mg cm$^{-2}$) pouch cells assembled with the quasi-solid electrolyte delivered highly stable electrochemical performances even at a high working temperature of 90 °C and after sustaining damage (171 mAh g$^{-1}$ after 300 cycles, 89% capacity retention; 164 mAh g$^{-1}$ after 100 cycles even after being bent and cut).

## Results and discussion

**Preparation of quasi-solid electrolyte via sub-nanoconfinement**. The CuBTC MOF with PSS polymer (poly(sodium 4-styrenesulfonate)) decorated inside its channels was employed as the host material (CuBTC-PSS, 6.5 Å) to fabricate a quasi-solid electrolyte (Fig. S1)[30]. Powder X-ray diffraction (XRD) measurements were first employed to primarily determine the presence of the liquid electrolyte confined inside the MOF channels (Fig. 2a, b). A reappearing (111) peak clearly suggested that there was a liquid electrolyte confined and coordinated inside the MOF channels, though this peak did not appear in the pattern of the activated MOF without any water molecules confined or coordinated inside its channels[21,31]. The apparently decreased pore size further suggested that the liquid electrolyte was successfully confined/coordinated inside the MOF channels (Fig. 2c). We defined the MOF-confined electrolyte as a quasi-solid electrolyte. Thermogravimetric analysis (TGA) was also employed to evaluate the thermal stability of the prepared quasi-solid electrolyte. The TG curve of a typical liquid electrolyte (1 M LiTFSI in propylene carbonate, abbreviated as 1 M LiTFSI-PC) showed two obvious weight losses: the first weight loss (highlighted in yellow) started at approximately 100 °C and can be ascribed to the decomposition of the liquid solvent, while the second (highlighted in light blue) was induced by the decomposition of lithium salt (LiTFSI) (Fig. 2d).

The TG curve of the quasi-solid electrolyte, however, demonstrated different results (Fig. 2e, f): Although two obvious weight losses were still clearly observed, the temperature at which the electrolyte began to lose weight was much higher, especially the decomposition temperature of the liquid solvent. The decomposition temperature (highlighted in yellow) of the liquid solvent within the quasi-solid electrolyte encountered the largest change: it began to decompose at nearly 200 °C, an improvement of almost 100 °C over that of the typical liquid electrolyte. The salt decomposition temperature (highlighted in light blue in both Fig. 2e, f) of the quasi-solid electrolyte also experienced an obvious increase, as it started at nearly 400 °C, almost 50 °C higher than that of a typical liquid electrolyte. The remarkably improved decomposition temperature can be ascribed to the unique effects of sub-nanoconfinement/coordination (physical confinement by narrow MOF channels and chemical interactions with metal sites inside the channels) of the porous polar MOF host[32–35] towards a tiny amount of liquid electrolyte. The effects of sub-nanoconfinement and coordination reported in this work are different from those reported in other works. Since most other studies reported sub-nanoconfinement and coordination in aqueous solutions[32–35], the sub-nanoconfinement and coordination demonstrated in this work focused on organic liquid electrolytes. Moreover, the most significant differences induced by the sub-nanoconfinement and coordination reported in this work are the aggregated electrolyte configurations and the largely improved decomposition temperature of the tiny amounts of liquid electrolyte inside the narrow MOF channels (these differences will be discussed in detail in the next section). We assumed that the unusual properties of the liquid electrolyte confined inside the MOF channels arise from the unique nature of the sub-nano porous MOF material, which exhibits both large-scale flexibility and polar heterogeneous internal surface (unsaturated polar Cu metal sides inside MOF channels enable strong physisorption or even chemisorption onto polar PC molecule solvents, which can be verified by the (111) peak shown in Fig. 2b), which attracts the polar liquid electrolyte solvent molecules used in this work. As schematically illustrated in Fig. 2g, because typical bulk liquid electrolytes do not exhibit physical confinement or coordination effects, they are prone to evaporation and thus possess relatively lower boiling points. Therefore, typical liquid electrolytes are generally unsafe, especially at high temperatures, and can cause LMBs to have potentially serious safety hazards, as shown in Fig. 1c[36]. In sharp contrast, due to the physical sub-nanoconfinement and chemical coordination with the MOF host (Fig. 2h), the quasi-solid electrolytes are unlikely to evaporate. Thus, the quasi-solid electrolyte prepared in this work possesses a much higher boiling point than its bulk liquid electrolyte counterpart and can be used more safely under high working temperatures.

**Physicochemical properties of the quasi-solid electrolyte**. The physiochemical properties of the prepared quasi-solid electrolyte were then further evaluated. First, the configuration of the quasi-solid electrolyte was tested by attenuated total reflectance Fourier transform infrared spectroscopy (ATR-FTIR) and Raman spectroscopy (Figs. 3a, b, S2). The tiny amount of liquid electrolyte confined inside the MOF channels demonstrated stronger Li-PC interactions and concentrated TFSI$^-$ than the typical diluent liquid electrolytes (1 M LiTFSI-PC, Fig. S3c and d), suggesting a much more aggregated electrolyte configuration. The tiny amount of liquid electrolyte confined inside the MOF channels was even more aggregated than the concentrated liquid electrolyte (3 M LiTFSI-PC, Fig. S3e, f). The electrochemical stability window of the prepared quasi-solid electrolyte was then evaluated by linear

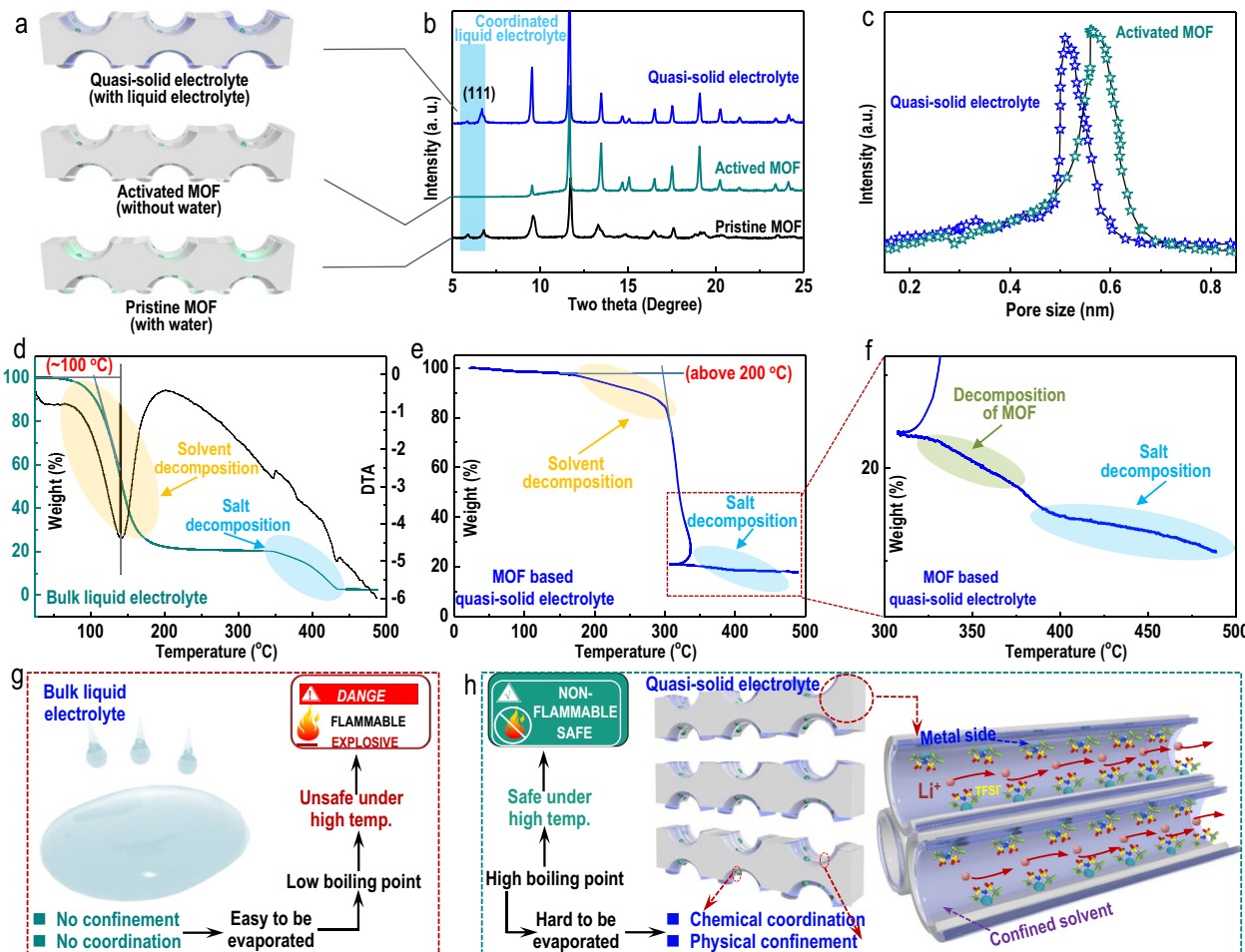

**Fig. 2 Physical characterization of the MOF-based quasi-solid electrolytes. a** Schematic illustration of the porous host material precursor used to prepare the electrolytes under different conditions. From bottom to top: pristine MOF (CuBTC-PSS), activated MOF, MOF with a liquid electrolyte coordinated inside its channels (quasi-solid electrolyte)) and the corresponding **b** X-ray powder diffraction (XRD) patterns of the three kinds of materials. **c** Pore size distributions of the activated MOF- and MOF-based quasi-solid electrolytes. **d** Thermogravimetric analysis (TGA) curve of the typical liquid electrolyte (1 M LiTFSI-PC). **e** TGA curve of the MOF-based quasi-solid electrolyte and **f** the enlarged version of that TGA curve (**e**). Schematic illustration of **g** the disadvantages of typical liquid electrolytes and **h** advantages of MOF-based quasi-solid electrolytes.

sweep voltammetry (LSV). The quasi-solid electrolyte exhibited an electrochemical stability window that was apparently extended to 5.4 V (blue curve in Fig. 3c), which was obviously much higher than that of typical liquid electrolytes (green curve in Fig. 3c). The potentiostatic intermittent titration (PITT) results presented in Fig. S3g and the cyclic voltammograms (CVs) presented in Fig. S3h further verified the excellent electrochemical stability of the prepared quasi-solid electrolyte. The excellent electrochemical stability of the quasi-solid electrolyte of the quasi-solid electrolyte can be ascribed to the sub-nanoconfinement/coordination effects of the MOF host towards the liquid electrolyte inside its channels and the aggregated electrolyte configuration formed inside its sub-nanochannels. To be more specific, a more aggregated electrolyte configuration usually indicates enhanced Li-PC solvents and Li-TFSI⁻ interactions. Due to the smaller but much more compact solvation sheaths of lithium ions, it was more difficult for the solvated PC solvents to be removed from the solvation sheaths of the solvated lithium ions and then undergo oxidation. Thus, compared with the dilute electrolyte, the quasi-solid electrolyte exhibited an apparently enhanced voltage window. Additionally, due to the sub-nanoconfinement effects promoted by the sub-nanochannels of the MOF, the tiny amount of liquid electrolyte confined inside the MOF channels decomposed at higher

temperatures than the typical liquid electrolyte (as shown in Figs. 2e, f). Therefore, more force is needed to oxidize and decompose the quasi-solid electrolyte than the typical liquid electrolyte. This is another reason for the high voltage window of the quasi-solid electrolyte. The thickness and quality of different electrolytes used in cell fabrication were also evaluated. The weight and thickness of the commercial $Li_{1.3}Al_{0.3}Ge_{1.7}(PO_4)_3$ (LAGP) solid-state electrolyte were found to be as high as 185.8 mg cm⁻² and 560 µm, with the assumption that no additional liquid electrolyte was used (most previous studies that used solid-state electrolytes still involved the addition of a certain amount of liquid electrolyte when assembling batteries) (Fig. S4). Cells assembled with a typical liquid electrolyte demonstrated a much lower electrolyte weight and separator thickness, which were 33.9 mg cm⁻² (1.5 mg cm⁻² for the PP separator; 32.4 mg cm⁻² for the added electrolyte) and 20 µm, respectively. Despite the greater thickness (38 µm), the quasi-solid electrolyte exhibited the lowest weight of merely 3.5 mg cm⁻² (with only 0.3 mg cm⁻² liquid electrolyte confined inside the MOF channels, corresponding to 0.23 µL cm⁻², as shown in Fig. S5). The apparently much lower electrolyte weight suggested the promising prospects of the quasi-solid electrolyte in constructing various high-energy-density LMBs. In addition to the significantly lighter weight, the

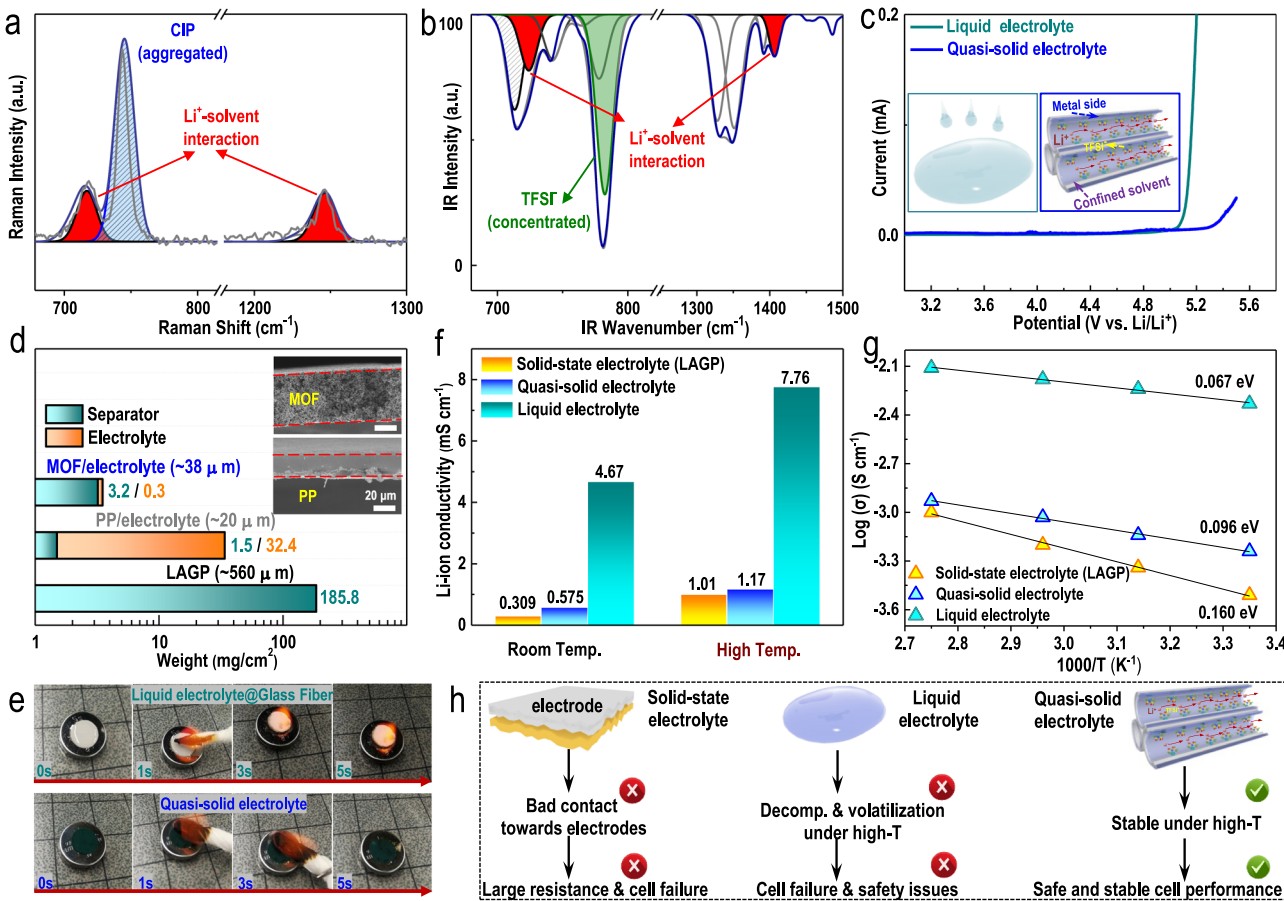

**Fig. 3 Physiochemical properties of the MOF-based quasi-solid electrolyte. a** Raman spectrum and **b** Fourier transform infrared spectrum (FT-IR) of the prepared MOF-based quasi-solid electrolyte. **c** Linear sweep voltammetry (LSV) curves of the two electrolytes (green curve: typical liquid electrolyte; blue curve: MOF-based quasi-solid electrolyte). The inset schematically illustrates the different electrolyte configurations. **d** Comparison of the thicknesses and qualities of different electrolytes (quasi-solid electrolyte, commercial PP separator typical liquid electrolyte and commercial $Li_{1.3}Al_{0.3}Ge_{1.7}(PO_4)_3$ (LAGP) solid electrolyte). Photographs of ignition tests of **e** typical liquid electrolyte saturated glass fibers and the prepared MOF-based quasi-solid electrolyte. **f** Ion conductivities and the corresponding activation energies **g** of the solid-state electrolyte, quasi-solid electrolyte and typical liquid electrolyte. **h** Advantages of the prepared quasi-solid electrolyte in constructing highly safe LMBs.

quasi-solid electrolyte is an excellent flame retardant (Fig. 3e). Compared with a typical electrolyte (directly dropped on glass fiber, upper panel, Fig. 3e), which can be ignited easily, the quasi-solid electrolyte, inspiringly, was totally nonflammable (bottom panel, Fig. 3e). This result suggested that the quasi-solid electrolyte was much safer than conventional liquid electrolytes. In addition, the lithium-ion conductivity of the prepared quasi-solid electrolyte was only slightly lower than that of the typical liquid electrolyte but obviously higher than that of the commercial LAGP solid electrolyte at both room temperature and a high temperature (Fig. 3f, left, 25 °C; right 90 °C). The activation energies (Ea) of the three electrolytes were also measured (Fig. 3g). The quasi-solid electrolyte demonstrated a much lower Ea than the commercial LAGP solid electrolyte (0.096 eV for the quasi-solid electrolyte, 0.160 eV for the commercial LAGP solid electrolyte). Though the typical liquid electrolyte exhibited the highest conductivity and lowest Ea, we still considered the quasi-solid electrolyte the most suitable electrolyte for constructing highly safe LMBs. As schematically demonstrated in Fig. 3h, the physical contact between the electrodes and solid-state electrolyte is generally extremely poor due to the stiffness of the electrolyte, which leads to high cell resistance and fast cell failure. Although typical liquid electrolytes exhibit the highest ion conductivity, volatilization and decomposition easily occur, especially at high

working temperatures, and these phenomena consequently cause cell failure and even create dangerous safety hazards. The promising quasi-solid electrolyte prepared in this work can not only enable a greatly improved electrode/electrolyte interphase than a solid-state electrolyte (due to its flexibility) but is also much more stable than typical liquid electrolytes, even at high temperatures. Therefore, the quasi-solid electrolyte showed promising prospects in constructing highly safe LMBs (Fig. S6).

**Compatibility of the quasi-solid electrolyte with the cathode and anode.** The compatibility of the prepared quasi-solid electrolyte with both the high-voltage cathode and lithium-metal anode was then investigated. First, the morphology of cycled $LiNi_{0.8}Co_{0.1}Mn_{0.1}O_2$ (NCM-811) cathodes harvested from $LiNi_{0.8}Co_{0.1}Mn_{0.1}O_2//Li$ (NCM-811//Li) half-cells assembled with either the typical liquid electrolyte or the prepared quasi-solid electrolyte were studied by scanning electron microscopy (SEM). The NCM-811 cathode cycled within a typical liquid electrolyte developed a rough surface that was covered by an uneven CEI layer after only 50 cycles (Fig. 4a, b). In sharp contrast, the NCM-811 cathode cycled within a quasi-solid electrolyte exhibited a totally different morphology: the NCM-811 particle maintained a smooth surface, and the CEI layer was barely observed even after

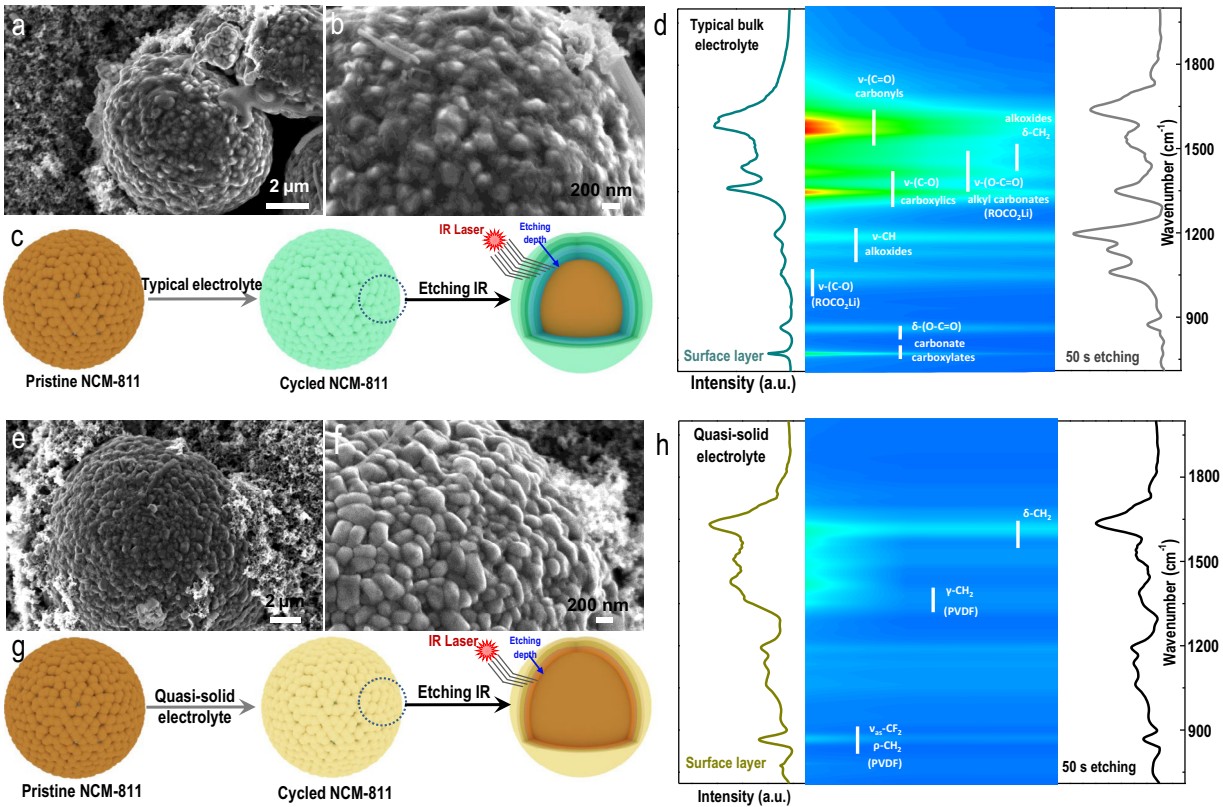

**Fig. 4 Characterizations of cycled NCM-811 cathode using both a typical electrolyte and the MOF-based quasi-solid electrolyte.** Scanning electron microscopy (SEM) images of the cycled NCM-811 cathode harvested from the NCM-811//Li half-cell assembled with **a**, **b** a typical liquid electrolyte (after 50 cycles) and **e**, **f** an MOF-based quasi-solid electrolyte (after 300 cycles). Argon-etching Fourier transform infrared spectrum (FT-IR) of the cycled NCM-811 cathode within **c** a typical liquid electrolyte (after 50 cycles) and **g** an MOF-based quasi-liquid electrolyte (after 300 cycles). Etching FT-IR spectra and the corresponding color mappings of the cycled NCM-811 cathode using the **d** typical liquid electrolyte and **h** MOF-based quasi-solid electrolyte.

300 cycles (Fig. 4e, f). However, the SEM cannot reveal detailed information about the thickness and components of the CEI layer covering the NCM-811 cathode. To collect more accurate results from the cycled NCM-811 cathode surface, high-resolution etching FT-IR spectroscopy was then employed (Fig. 4c, g, 50 s of argon etching with 2 s as an interval; the total etching depth was ~50 nm). For the cycled NCM-811 cathode harvested from the cell using a typical liquid electrolyte (Figs. 4d and S7b), the mapping image demonstrated numerous remarkable byproduct-related peaks that were induced by side reactions that occurred during the electrochemical cycling processes. To be more specific, strong peaks located at ~1355 and 1596 cm$^{-1}$, which can be assigned to carboxyl groups (C–O) and carbonyl groups (C=O), respectively, were clearly observed. These two products mainly originated from the decomposition of the liquid electrolyte (PC solvent)[22]. These two characteristic peaks were detected throughout almost the entire argon etching process, which enabled direct quantitative analysis of the thickness (50 nm in total) of the CEI layer. In stark contrast, only peaks related to the PVDF binder were detected throughout the entire argon etching process when the cycled NCM-811 cathode harvested from the cell using a quasi-solid electrolyte was used (Figs. 4h and S7a). The corresponding results from the high-resolution etching XPS (Fig. S7c and d) are consistent with the etching FT-IR results shown in Fig. 4c. This result indicates that after the quasi-solid electrolyte was used, the NCM-811 cathode experienced significantly reduced electrolyte decomposition and a greatly improved electrolyte/cathode interphase (CEI-free cathode

surface). We ascribed this observation to the aggregated configuration of the quasi-solid electrolyte induced by sub-nanoconfinement in the MOF channels. Due to the enhanced Li-PC solvents and Li-TFSI$^-$ interactions, the solvation sheaths of the lithium ions decreased but became much more compact. The number of PC solvent molecules contained within each solvation sheath of the solvated lithium ion (a tiny liquid electrolyte confined inside MOF channels, containing only strongly coordinated PC solvents, as shown in Fig. 3a) was much lower than those of both the dilute electrolyte (which contained various free PC solvents and weakly coordinated PC solvents within the solvated lithium ions, as shown in Fig. S3c) and the concentrated electrolyte (which contained both various weakly coordinated and strongly coordinated PC solvents within the solvation sheaths of the solvated lithium ions, as shown in Fig. S3e). Benefiting from the eliminated free PC solvents, the weakly coordinated PC solvents and some of the strongly coordinated PC solvents within solvated lithium ions of the quasi-solid electrolyte, the number of byproducts of the decomposition of the PC solvent molecules was greatly reduced, thus leading to a nearly CEI-free NCM-811 cathode.

Its compatibility with highly reactive lithium-metal anodes was consequently evaluated by Li//Li symmetrical cells. The superb Li//Li symmetric cell performance suggested good reversibility of Li plating and stripping processes benefitting from the employment of the prepared quasi-solid electrolyte (Fig. S8). To further support this conclusion, the electrochemical performances of LiFePO$_4$//Li (LFP//Li), Li$_4$Ti$_5$O$_{12}$//Li (LTO//Li), and NCM-811//

Li half-cells assembled with the prepared quasi-solid electrolyte were also investigated. The excellent electrochemical stabilities of all cells further suggested that the prepared quasi-solid electrolyte was compatible with the lithium-metal anode (LFP//Li, Fig. S9a and b; LTO//Li, Fig. S9c; NCM-811//Li, Figs. S10 and S11). SEM images of the cycled Li from the Li//Li symmetrical cells assembled with either a typical liquid electrolyte or the prepared quasi-solid electrolyte were recorded. It is widely acknowledged that conventional carbonate-based electrolytes are incompatible with lithium metal[12]. Generally, LMBs assembled with typical carbonate-based electrolytes have limited lifespans due to the rapid electrolyte decomposition and uncontrolled growth of dendritic lithium occurring on the lithium metal surface[7,12,19]. Consistent with other reported studies, the cycled lithium metal harvested from Li//Li half-cells assembled with a typical liquid electrolyte exhibited a rough surface on which numerous unevenly distributed lithium dendrites could be clearly observed (Fig. S12a, b). The cycled lithium-metal harvested from the Li//Li half-cells assembled with quasi-solid electrolyte presented a very smooth surface, while very little dendritic lithium was observed (Fig. S12c, d). This result indicates that the poor compatibility of conventional liquid carbonate-based electrolytes with lithium-metal has been perfectly solved by using the prepared quasi-solid electrolyte. The remarkably enhanced electrolyte/lithium-metal interphase can be ascribed to the unique aggregated electrolyte configuration and consequently remarkably reduced contact opportunity between electrolyte solvents and reactive lithium-metal, which resulted from the tiny amount of liquid electrolyte contained within the prepared quasi-solid electrolyte.

These characterizations only focused on the surface of the NCM-811 cathode. To further understand the working mechanism of the prepared MOF-based quasi-solid electrolyte in lithium-metal batteries, detailed information from inside the depths of the cycled cathode needs to be further studied. Therefore, the cycled NCM-811 cathode collected from NCM-811//Li using a quasi-solid electrolyte (after 700 cycles, harvested from the cell demonstrated in Fig. S11) was studied by Raman spectroscopy. As schematically demonstrated in Fig. 5a, to collect detailed and accurate information even inside the depths of the cycled NCM-811 cathode, a unique tape peeling test was used to peel off the surface layers of the cycled NCM-811 cathode and thus expose the new NCM-811 cathode interphases to the Raman laser. After each tap peeling (from 0 to 9 times), new interphases of the NCM-811 cathode were observed at different depths (different thicknesses, Fig. 5b–g). Then, the obtained Raman spectra from each depth were further investigated. As demonstrated in Fig. 5h, two apparent peaks related to the liquid electrolyte were constantly detected at all depths. Moreover, the two peaks maintained almost the same shapes as the shapes of the peaks corresponding to the liquid electrolyte confined inside the MOF channels of the quasi-solid electrolyte. This result suggested that the liquid electrolyte can exit the MOF channels of the quasi-solid electrolyte and consequently wet the NCM-811 cathode deep into the interior despite the gradual decrease in the intensities of the electrolyte-related peaks (Fig. 5i). In addition, as schematically illustrated in Fig. S13a, two types of liquid electrolytes were confined inside the MOF channels: one was the liquid electrolyte under only physical confinement (type 1), and the other was the liquid electrolyte under both physical and chemical confinement (type 2). We hypothesized that the physically confined liquid electrolyte (type 1) exited the MOF channels and wetted the cathode (as schematically demonstrated in Figs. S13b and S14a). The gradually increasing BET results shown in Fig. S14b indicated the constant consumption of the liquid electrolyte. Apparent (111) peaks (indicating the chemically bonded liquid electrolyte), were still observed in the XRD patterns of the cycled

MOF-based quasi-solid electrolytes (Fig. 5j), which suggests that the liquid electrolyte under both physical and chemical confinement (type 2) cannot exit the MOF channels to wet the cathode. Therefore, the constant consumption of the liquid electrolyte could only be ascribed to the liquid electrolyte (type 1) physically confined inside the MOF channels. Additionally, as schematically illustrated in Fig. S13c, since both physically and chemically confined liquid electrolytes (type 2) remained inside the MOF channels, sub-nanoconfinement constantly occurred throughout the electrochemical cycling process.

**Electrochemical performance of quasi-solid electrolyte-powered NCM-811//Li pouch cells under harsh conditions.** The prepared quasi-solid electrolyte demonstrated an extended electrochemical stability window, largely enhanced interfacial properties, remarkably suppressed electrolyte decomposition, and significantly eliminated dendritic lithium formation during cycling. More importantly, benefiting from the unique electrolyte fabrication strategy, the prepared quasi-solid electrolyte also exhibited a high boiling point, a much-improved decomposition temperature, and the potential for safe operation even at high working temperatures. To further verify our conjecture, the electrochemical performance of NCM-811//Li pouch cells assembled with the prepared quasi-solid electrolyte was tested at both room temperature (25 °C) and a high temperature (90 °C). An NCM-811//Li pouch cell using a typical liquid electrolyte was also fabricated and measured for comparison. A quasi-solid electrolyte-based NCM-811//Li pouch cell was fabricated, as shown in Fig. 6a (pouch-cell size: $4 \times 5$ cm$^2$). After successful assembly, the quasi-solid electrolyte-based NCM-811//Li pouch cell (with a high NCM-811 mass loading of 20 mg cm$^{-2}$) cycled at room temperature demonstrated excellent cycling performance (Fig. 6c, blue curve). When tested under harsh conditions at 90 °C, surprisingly, the quasi-solid electrolyte-based NCM-811//Li pouch cell still delivered a high initial capacity (191.5 mAh g$^{-1}$) and ultra-stable cycling stability (300 cycles, capacity sustained at 171.2 mAh g$^{-1}$, corresponding to almost 90% capacity retention), as shown in Fig. 6b, c (yellow curve). To the best of our knowledge, this result is the best pouch-cell performance obtained at such a high working temperature[6,9,13,37,38]. The NCM-811//Li pouch cell assembled with a typical liquid electrolyte, however, demonstrated very poor electrochemical performance at both normal room temperature (Fig. 6d; blue curve in Fig. 6e) and high temperature (yellow curve in Fig. 6e). When at room temperature, it experienced fast capacity decay after only 20 cycles and was sustained at only 19.9 mAh g$^{-1}$ after 32 cycles. When cycled at high temperature, the NCM-811//Li pouch cell assembled with a typical liquid electrolyte delivered a much worse performance: it could only cycle for 30 cycles and underwent fast capacity decay after 5 cycles, which was finally sustained at 22.7 mAh g$^{-1}$. To simulate the practical battery working conditions, the quasi-solid electrolyte-based NCM-811//Li pouch cell was then tested under even harsher conditions: both a high working temperature (90 °C) and sustaining damage (bent and cut). If the pouch cell could work normally under such severe conditions, then the use of a quasi-solid electrolyte would represent a drastic technological leap towards the practical utilization of various secondary batteries. To our surprise, the NCM-811//Li pouch cell assembled with the prepared quasi-solid electrolyte demonstrated an impressive electrochemical performance even after taking damage (bent and cut) and at 90 °C (Fig. 6f). Even after being bent and at a high temperature (90 °C), the NCM-811//Li pouch cell assembled with the quasi-solid electrolyte delivered almost the same capacity (although there was apparent capacity decay after the pouch cell was bent in the initial several cycles, and the capacity of

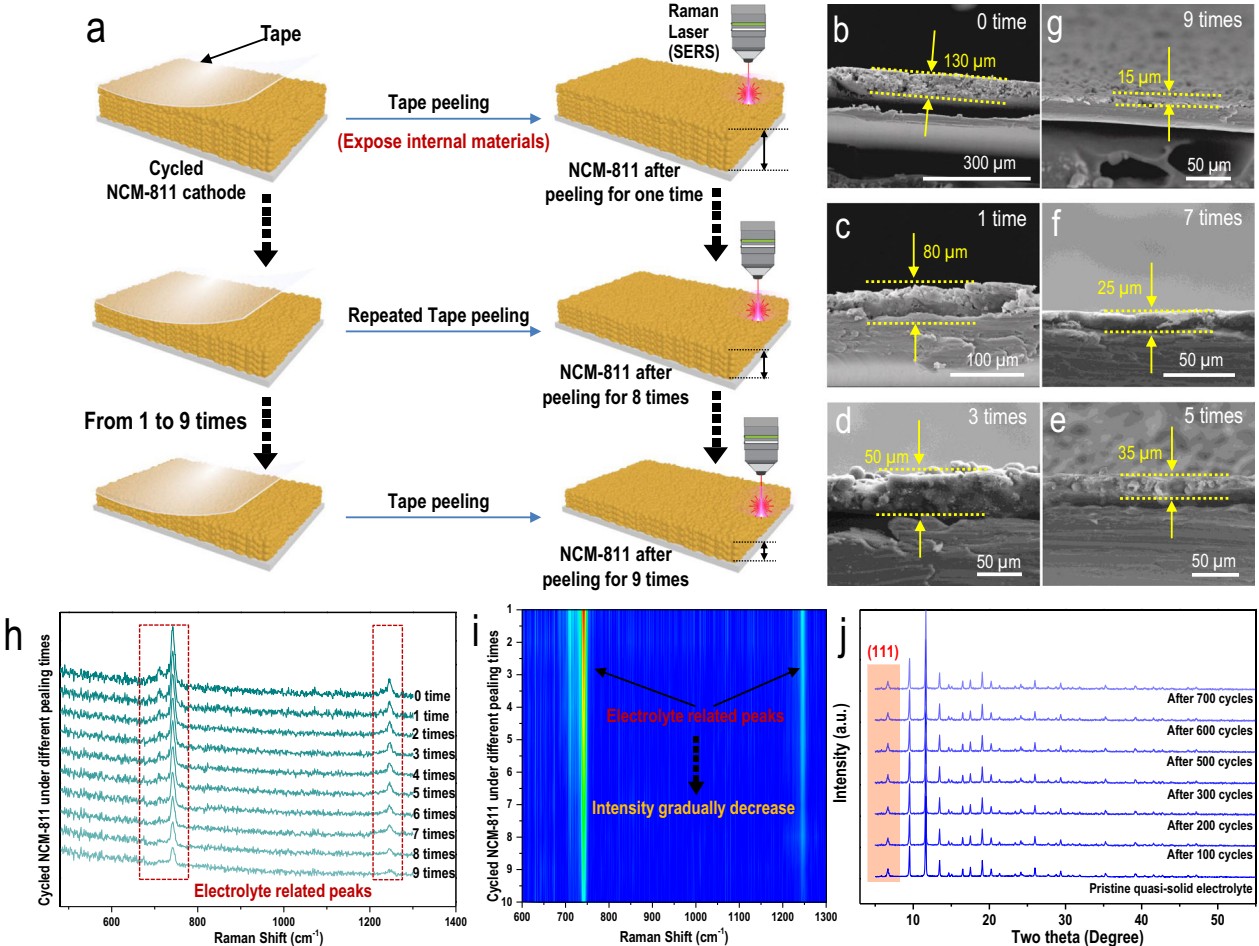

**Fig. 5 Characterizations of the cycled quasi-solid electrolyte and cycled NCM-811 cathode at different depths. a** Schematic illustration of the tape peeling method employed to peel off certain amounts of NCM-811 materials and expose the internal area of the cycled NCM-811 cathode for the following Raman experiment (after 700 cycles). SEM images of the cycled NCM-811 cathode under tape peeling tests for **b** 0 times, **c** 1 time, **d** 3 times, **e** 5 times, **f** 7 times and **g** 9 times. **h** Raman spectra detected from the cycled NCM-811 cathode under different times of tape peeling test (from 0 to 9 times) and the corresponding **i** color mapping. **j** XRD patterns of pristine and cycled quasi-solid electrolytes.

the pouch cell was consequently sustained at approximately 167.7 mAh g$^{-1}$, 36–77 cycles) as that before the cell was bent (1–35 cycles, the capacity stabilized at ~170.5 mAh g$^{-1}$). Even after being cut in the 78th cycle, the NCM-811//Li pouch-cell assembled with quasi-solid electrolyte maintained a high capacity of 166.2 mAh g$^{-1}$ (78–100 cycles), succumbing to slight capacity decay from the previous several cycles. A NCM-811//Li pouch cell with a typical liquid electrolyte was also assembled and tested after being bent and cut. However, due to the fast capacity decay of this cell at a high temperature, we only measured it at room temperature (25 °C) (Fig. S15). During the first seven cycles, the pouch cell delivered a high capacity of ~186.3 mAh g$^{-1}$. After being bent, the capacity dropped suddenly (143.7 mAh g$^{-1}$) and nearly reached zero very quickly after being cut. The SEM images of the cycled NCM-811 cathodes from both the typical liquid electrolyte (Fig. S16a and b) and the quasi-solid electrolyte (Fig. S16c and d) demonstrated similar morphologies, as shown in Fig. 4a b, e, f, respectively. The extraordinary performances of the pouch cells harvested under such harsh conditions (being damaged and testing at 90 °C high temperature) suggested the significant importance of using the prepared quasi-solid electrolyte when constructing highly safe and efficient LMBs. Benefitting from the eliminated free PC solvents, weakly coordinated PC solvents and some of the strongly coordinated PC solvents containing

the solvated lithium ions of the quasi-solid electrolyte (aggregated electrolyte configuration), the byproducts of the decomposition of the PC solvent molecules were greatly reduced. Because there were much fewer side reactions, the pouch cells assembled with quasi-solid electrolyte demonstrated good cycling performances. The greatly improved decomposition temperature and the quasi-solid property of the flexible quasi-solid electrolyte made the pouch-cell work stably and safely even in harsh environments, including at a high temperature (90 °C) and after taking damage (bent and cut). Moreover, the SEM images and the corresponding XRD results shown in Fig. S17 suggested the excellent stability of the quasi-solid electrolyte even after various electrochemical cycling processes were performed. These obtained results together suggest the promising potential of the quasi-solid electrolyte for promoting the long-term life of batteries.

Unlike from traditional LIBs/LMBs, which can only work safely under mild conditions, such as room temperature (25 °C) and undamaged conditions, the highly safe and stable NCM-811//Li pouch cells created in this work by using a stable quasi-solid electrolyte were effective even under the harsh working conditions of extremely high temperature (90 °C) and damage (bent and cut). Specifically, NCM-811//Li (NCM-811 mass loading of 20 mg cm$^{-2}$) pouch cells assembled with the quasi-solid electrolyte delivered highly stable electrochemical

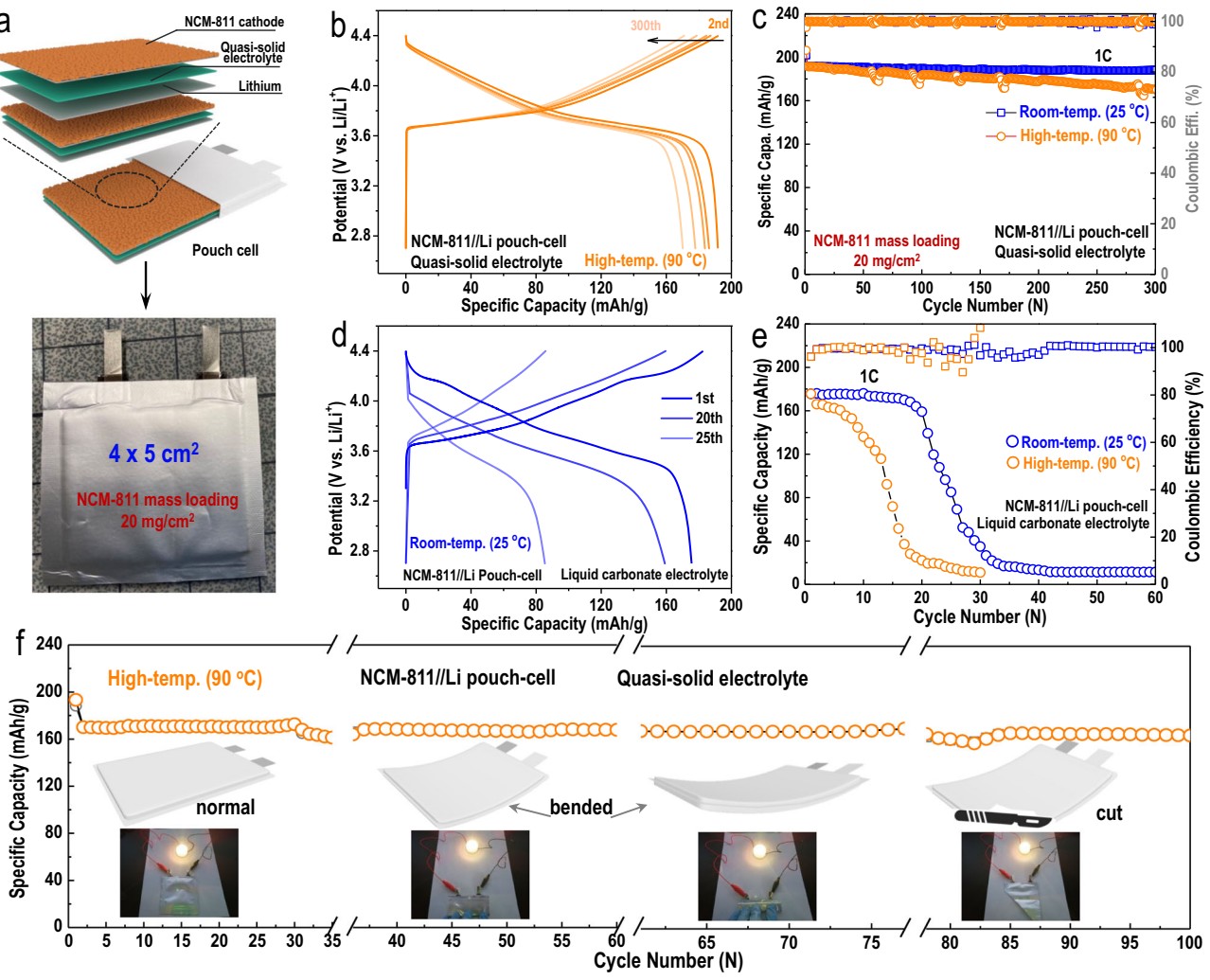

**Fig. 6 NCM-811//Li pouch cells used a quasi-solid electrolyte under harsh working conditions. a** Schematic illustration of the cell structure and digital photo of an NCM-811//Li pouch cell assembled with a MOF-based quasi-solid electrolyte. **b** Discharge/charge curves and **c** the corresponding cycling performances of the NCM-811//Li pouch cell assembled with a quasi-solid electrolyte under a high cathode mass loading (approximately 20 mg cm$^{-2}$) and at high temperature of 90 °C (yellow curve). The blue curve represents the cycling performance obtained at room temperature (25 °C). **d** Discharge/ charge curves and **e** the corresponding cycling performances of the NCM-811//Li pouch cell assembled with a typical liquid electrolyte under a high cathode mass loading (~20 mg cm$^{-2}$) and room temperature of 25 °C (blue curve). The yellow curve represents the cycling performance obtained at a high temperature of 90 °C. **f** Cycling performance of an NCM-811//Li pouch cell (NCM-811 cathode mass loading approximately 20 mg cm$^{-2}$) assembled with a MOF-based quasi-solid electrolyte under a harsh working environment of high working temperature (90 °C) and after sustaining damage (bent and cut).

performances even at a high working temperature of 90 °C (171 mAh g$^{-1}$ after 300 cycles, 89% capacity retention; 164 mAh g$^{-1}$ after 100 cycles even after being damaged; this damage included bending and cutting). The nonflammability of the quasi-solid electrolyte originated from the high boiling point induced by both the physical confinement and chemical coordination of the tiny amount of liquid solvent inside the channels with the host material (extremely low loading < 0.23 μL cm$^{-2}$, equal to 0.3 mg cm$^{-2}$). Moreover, due to the aggregation of the tiny amount of liquid electrolyte contained within the host material, the quasi-solid electrolyte exhibited decent lithium-ion conductivity and an extended electrochemical voltage window (approximately 5.4 volts versus Li/Li$^+$). Through the use of a quasi-solid electrolyte, the long-lasting and difficult safety challenges preventing LMBs from being used at high temperatures have been successfully addressed. We anticipate that the quasi-solid electrolyte reported in this work will speed up the practical application of ultrasafe and highly efficient LIBs/LMBs and

ultimately power our society in a safer way, even under extreme conditions.

## Methods

**Materials and methods**. All the chemicals employed in this synthesis section were purchased from Wako Pure Chemical Industries Ltd. without additional exception.

### Preparation of different MOF composites

*Preparation of the PSS modified MOF composites.* (1) Synthesis of copper hydroxide nanorods (CHNs): Copper hydroxide nanostrands (CHNs) were firstly synthesized by quickly mixing equal volume 2.32 g copper nitrate hexahydrate solution (600 ml) with 200 mg aminoethanol aqueous solution (600 ml) at room temperature and aged for 48 h. The CHNs composites were collected by filtering the mixture onto an organic membrane. Then, the CuBTC MOF was successfully obtained by immersing the CHNs film into 20 mM Benzene-1,3,5-tricarboxylic acid (BTC) water-ethanol (water/ethanol volume ratio 1:1, 40 ml) solution at room temperature for 12 h. The obtained composite was finally vacuumed at 180 °C for 72 h to generate the activated MOF sample.

(2) Synthesis of PSS modified MOF (CuBTC-PSS): After the CHNs were obtained, 24 mg PSS (Poly(sodium 4-styrenesulfonate)) was added into 600 ml

CHNs solution to prepare the CHNs-PSS composites. Then the CuBTC-PSS MOF was successfully obtained by immersing the CHNs-PSS film into 20 mM Benzene-1,3,5-tricarboxylic acid (BTC) water-ethanol (water/ethanol volume ratio 1:1, 80 ml) solution at room temperature for 12 h. The obtained composite was finally vacuumed at 180 °C for 72 h to generate the activated MOF-PSS sample[30].

**Preparation of MOF film**. MOF based solution was prepared by thoroughly mixing 90 wt. % activated MOF sample with 10 wt% Polytetrafluoroethylene (PTFE) in ethanol. The obtained MOF slurries were then quickly stirred to mix the sample evenly and evaporate the unnecessary organic solvent, finally resulting in sticky MOF putty. The obtained MOF putty was then uniformly spread on the Al foil and then dried at 80 °C for 10 min in drying oven. The MOF coated Al foil was then immersed into methanol for 5 min until the MOF film was detached from the Al foil and formed the flexible MOF film[39]. The MOF film was then further compacted more than 10 tons of pressure to remove the inter-particle pores. The obtained MOF film was firstly dried at 80 °C for 1 h in drying oven and then followed by vacuumed dried oven at 180 °C overnight to activate the MOF film. The prepared activated MOF film was then cut into small plates (16 mm in diameter) and re-activated under vacuum at 180 °C overnight before transferred into glove box for further usage.

**Preparation of carbonate electrolyte**. Before this process, carbonate electrolytes (PC-LiTFSI) with different concentrations (1 M and 3 M) were firstly prepared. Typically, for the preparation of 1 M PC-LiTFSI carbonate electrolyte, 2.87 g LiTFSI salt was mixed with 10 mL Propylene Carbonate (PC) solvent and stirred at 60 °C for 2 h. Concentrated carbonate electrolytes (3 M PC-LiTFSI) were prepared follow the same procedures except stoichiometric LiTFSI salt was added.

**Preparation of quasi-solid electrolyte**. In this work, we selected 1 M LiTFSI-PC as the typical liquid electrolyte for comparison. To prepare the quasi-solid electrolyte, we harvested the cycled MOF films from the surface of the cycled Li-metals in Li//Li cells after 10 cycles and defined the electrolyte confined inside the channels of MOF as the quasi-solid electrolyte. After that, the quasi-solid electrolyte was finally obtained and can be used to fabricate various batteries in this work.

**Electrodes preparation**. The obtained LiNi$_{0.8}$Co$_{0.1}$Mn$_{0.1}$O$_2$ (defined as NCM-811, provided by Prof An-Min Cao from Chinese Academy of Sciences (CAS)), and lithium foil (Lion Chemical Industry Co., Ltd.) were employed as electrode materials. Generally, 1.0 g electrode powders mixed with carbon black and poly-vinylidene fluoride (PVDF, Du Pont-Mitsui Fluorochemicals Co. Ltd.) powder in a ratio of 8:1:1 and then directly stirring for 4 h to get a viscous solution. The obtained slurry was then homogeneously coated onto Al foil current collector by a scraper. After tiny pressing procedure, the active materials-loaded Al foil was vacuum dried at 110 °C overnight. The mass loading of the NCM-811 cathode materials was about 20 mg/cm². Part of the obtained NCM-811 cathode was cut into final electrode plates (11 mm in diameter) for coin-cell fabrications, while the rest of the obtained cathode was cut into a rectangle (4 × 5 cm² in size) for pouch-cell fabrications. LiFePO$_4$ (LFP) and Li$_4$Ti$_5$O$_{12}$ (LTO) based electrodes were also prepared following the similar procedure. And the two obtained electrodes were cut into final electrode plates (11 mm in diameter) for coin-cell fabrications. The mass loading of both LFP and LTO were about 6 mg/cm².

**Cell assembly and electrochemical measurements**. CR2032 coin cells were assembled in an argon-filled glove box, in which both the moisture and oxygen contents were controlled to be less than 1 ppm. The prepared quasi-solid electrolytes were closely attached to the cathodes and followed by a physical pressing process. The obtained cathodes attached with quasi-solid electrolytes were then physically pressed on the surface of Li anodes for cell assembling. For comparisons, cells using typical liquid electrolyte carbonate electrolytes (1 M PC-LiTFSI, 70 μL) were also assembled accompanied with the glass fiber as separators. The NCM-811//Li coin-cells and pouch-cells were operated with a potential limit between: 2.7–4.4 V in the study. Before each electrochemical characterization, the cells were kept on open circuit for 10 h. All of the potentials in this study were referenced to Li/Li+. The galvanostatic electrochemical measurements were carried out under potential control using the battery tester system HJ1001SD8 (Hokuto Denko) at both room temperature (25 °C) and high temperature (90 °C). For the Linear sweep voltammetry (LSV) and EIS tests, the electrochemical experiments are carried out under the control of a potentiostat (Potentiostat/Galvanostat PGSTAT30, Autolab Co. Ltd., Netherlands). The current and potential outputs from the potentiostat were recorded by a multifunction data acquisition module/amplifier (PGSTAT30 Differential Electrometer, Autolab), which was controlled by General Purpose Electrochemical Software (GPES). The ionic conductivity was measured by a symmetric coin cell with two stainless steel electrodes. To test the ionic conductivity of the typical liquid electrolyte (1 M LiTFSI-PC), Glass Fiber (GF) wetted by electrolyte was sandwiched between two stainless steel sheets. The quasi-solid electrolyte was sandwiched between two stainless steel sheets. Noted that to make sure the quasi-solid electrolyte was successfully formed, we harvested the MOF layer with quasi-solid electrolyte from a cycled (10 cycles) quasi-solid electrolyte

used Li//Li half-cell by physically peel off the MOF layer coated on the electrode. After the quasi-solid electrolyte was successfully obtained, it is hence can be used as membrane to sandwiched between two stainless steel sheets for ionic conductivity test.

## Morphology and structure characterization

*SEM, XRD, BET and XPS characterizations.* The morphology of the as-prepared MOF products, pristine cathodes and cycled cathodes and Li anodes were characterized with scanning electron microscopy (SEM, JEOL JSM-6380LV FE-SEM). X-ray diffraction (XRD) measurements were performed on a Bruker D8 Advanced diffractometer fitted with Cu-Kα X-rays (λ = 1.5406 Å) radiation at a scan rate of 0.016°/s. For the pre-treatment procedures: The cycled cells were transferred into an Ar glove box once the electrochemical treatments were finished, and the electrodes were extracted from the cell and placed in a glass bottle. The electrode plates were twice rinsed by dimethoxyethane (DME, Sigma Aldrich, 99%) to wash off the electrolyte salt and the residual solvent, and then evaporated in a vacuum chamber, connected to the glove box, for 12 h. The dried electrode plates were moved back to glove box and placed onto a SEM sample holder. The sample holder was sealed in an airtight container and then transferred into the SEM sample loading chamber. Note that, in order to restrain the exposure time to the ambient, samples (cycled electrode plates) were tightly sealed into a glass bottle (fill with Ar gas), and transferred to the related chambers (SEM) as quickly as possible. Thus, we assumed the morphology and the component of electrode surface would not obviously change for such a short time exposure to the open air. The specific surface area (SSA) the cycled MOF-based quasi-solid electrolytes after different cycles was determined based on Brunauer-Emmett-Teller (BET) theory in the relative pressure range of 0.04–0.2.

*Spatial resolution Operando-Raman spectroscopy characterizations.* The Raman spectra were recorded using a JASCO microscope spectrometer (NRS-1000DT). The spectral resolution of the Raman spectra in the study was ca. 1.0 cm⁻¹. Typically, the scattering signal in Raman spectrum was weak and hard to be investigated. In this case, in order to obtain strong and clear peaks on the spectra, we took advantage of a shell-isolated nanoparticle-enhanced Raman spectroscopy (SHINERS) technique that evidently enhances the scattering signal[40,41]. Briefly, Au nanoparticles (NSp) were synthesized with a diameter of 30–40 nm as core by a standard sodium citrate reduction method. Then freshly prepared aqueous solution of 1 mM (3-aminopropyl) trimethoxysilane (APS, Sigma Aldrich) was added to the gold sol under vigorous magnetic stirring for 15 min, followed by the addition of a 0.54 wt % sodium silicate solution (Tokyo Chemical Industry Co., Ltd). Then, the solution was heated to 90 °C under vigorous magnetic stirring for 1 h. The series of steps ensures the formation of an ultra-thin SiO₂ shell (2–4 nm) without any pinhole. The washed and dried Au@SiO₂ NSp were re-dispersed in ethyl alcohol. Finally, the obtained Au NSp solution was mixed with MOF particles together with NMP as solvent before the NSp contained MOF was coated on the surface of NCM-811 cathode for Raman experiment. These Raman samples were further dried in vacuum at 80 °C for 18 h before assembled into the cell. Note that the amount of deposited NSp was very small, so that we assumed it would not cause any influence on the electrochemical behaviors.

*Operando-Raman spectroscopy characterization for cycled NCM-811 cathode.* To verify the NCM-811 cathode can be wetted by liquid electrolyte came out from the MOF channels, the Raman spectra of cycled NCM-811 cathode under different depths after different times of Tap peeling test were collected. By constantly peeling off the surface layer of cycled NCM-811 cathode, the internal sections in deep depths of NCM-811 can be exposed. Then, Raman measurement was applied to detect whether there were liquid electrolyte peaks contained inside the deep depth of the cycled NCM-811 cathode.

## FT-IR characterizations

*Attenuated total reflection Fourier-transform infrared (ATR-FTIR) characterizations.* ATR-FTIR measurements were carried out on a FT/IR-6200 spectrometer (JASCO Corp.) coupled with Platinum Diamond ATR, which consists of a diamond disc as an internal reflection element. The typical electrolyte and MOF-based quasi-solid electrolyte were placed on the ATR crystal, and then the spectrum was recorded.

**Etching Fourier-transform infrared (FTIR) characterization**. For pretreatment, the cycled cells were transferred into an Ar glove box once the discharge finished, and the cathodes were extracted from the cell and twice rinsed by dimethoxyethane (DME, Sigma Aldrich, 99%) to wash off the electrolyte salt and the residual solvent, and then evaporated in a vacuum chamber, connected to the glove box, for ~45 min. The washed cycled cathodes were attached on the plate of etching IR instrument. And the IR spectra were collected with 2 sec as interval. As a result, 25 points were recorded. The cycled NCM-811 cathode used both typical liquid electrolyte and quasi-solid electrolyte were tested using the etching FT-IR following the aforementioned experiment processes.

*Depth-resolution etching X-ray photoelectron spectroscopy (XPS) characterization.* XPS measurement was performed using a VG scientific ESCALAB 250 spectrometers with monochromic Al Kα Ka source (1486.6 eV) under ultra-high vacuum. Similarly, to prevent long-time exposure to air environment, the samples (after rinsing and dry) were tightly sealed into an Ar-filled bottle and then soon transferred into XPS chamber as quickly as possible. The XPS was equipped with etching with different depth to analysis the component distribution.

## Data availability

All data generated in this study are provided in the Source data file and its Supplementary Information. Source data are provided with this paper.

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

## Acknowledgements

The authors thank Prof. An-Min Cao from the Key Laboratory of Molecular Nanostructure and Nanotechnology, Beijing National Laboratory for Molecular Sciences, Institute of Chemistry, Chinese Academy of Sciences (CAS) for providing NCM-811 cathode materials. We also thank Dr. S.J. Li from the National Institute for Materials Science (NIMS) for her valuable advice in preparing this paper. This research was partially supported by the National Natural Science Foundation (NSF) of China (21673166, 21633003 and U1801251, H.Z.). X.Z. acknowledges scholarships from the China Scholarship Council (CSC).

## Author contributions

Z.C. and H.Z. led the design of the study. Data collection and analysis were conducted by Z.C. The SEM and XRD measurements were performed by H.Y. and X Z., and P.H. helped with the XPS measurements. All authors co-wrote the manuscript. H.Z. supervised the work. All authors discussed the results and commented on the manuscript.

## Competing interests

The authors declare no competing interests.
