## [Peer Review File · Nature Communications]

REVIEWER COMMENTS

Reviewer #1 (Remarks to the Author):

The manuscript by Chang et al. demonstrated a quasi-solid electrolyte by using a microporous metal-organic framework (MOF) with a small amount of liquid electrolyte. The integration of physical nanoconfinement effect and chemical coordination endows a strong interaction of MOF skeleton and electrolytes, which is desirable for the improvement of the high-temperature physicochemical properties. In addition, the nanopores of MOFs seems to also render highly ion-pairing solvation, which was found to be correlated to a stable cycling performance at both room and elevated temperatures. On the whole, the manuscript reads interesting and related characterization data have been collected to support their proposed concept. However, some of the descriptions on the experimental results as well as the discussion raise a number of questions that prevents the acceptance of this manuscript in its current state. I would suggest the suitable revisions to address the following concerns:

1. It might be unreasonable to say an excellent conductivity in line 22, because the MOF trapped electrolyte shows noticeably reduced value compared with the bulk liquid as shown in Figure 3f.

2. In line 103-107, the total mass of liquid plus MOF membrane and liquid plus the control separator should also be pointed out.

Also, it is unfair to compare the amount of electrolyte in coin cells in line 105-107 to showcase the advantage of the quasi-solid electrolyte as the coin cell assembly cannot be scaled up directly to commercial cells with lean electrolyte.

3. The sub-nanoconfinement effects are the basis of this manuscript and have been mentioned several times while no references or evidence to explain and support these concepts in the current liquid PC/MOF system. For the nanoconfinement of electrolytes for batteries, a recent paper has demonstrated similar phenomena in gaseous system (Nat. Comm., 2021, 12, 3395). In addition, some comments are necessary to be added to highlight the difference in the concept as well as fundamentally address how the confinement occurs from molecular point of view.

4. It is hard to understand what is the coordination effect and how it works to endow a strong interaction between MOFs and electrolytes. Is it specific chemistry related? If PC is changed to other solvent, what will happen? In other words, the nanoconfinement should be a general phenomenon and needs to be demonstrated in a broader perspective.

5. Why the quasi-solid electrolyte film has to be made by Li/Li symmetric cell and cycling for 10 cycles and the harvested from the cycled cells? How realistic is this method to be adopted in any practical cell fabrication application?

6. Is any additional liquid electrolyte dropped in the cathodes? How the Li⁺ diffusion in the cathode? The experiment part describes the cell assembly by attaching a quasi-solid electrolyte to a cathode film, it could be possible to the cathode particles that in direct contact with the quasi-solid electrolyte layer to be redox active, but the particles inside the cathode that are far away from the electrolyte layer has no direct contact, how the ion transport occurs?

1) If pure solid-state diffusion, the kinetics will be super low for 20mg/cm². The authors should provide rate performance test show evaluate.

Or 2) if the ion transport is supported by liquid, then the nanoconfinement effect may disappear/ be invalid through the cathode layer.

Or if there is any other way?

7. Any comments on the selection of MOFs and polymers for the construction of MOF-based porous hosts? Any comment on the polytetrafluoroethylene (PTFE) as the polymer binder? In addition, what are the purposes/functions of the polytetrafluoroethylene (PSS) in this work?

8. Any comment on the long-term effects or on the projected shelf life of the batteries with the quasi-solid electrolyte?

9. What is the scientific value of fig 1? It's more like something designed for a review paper.

10. It is surprising that Cu-based MOF can be stable in such wide voltage window. CA experiment with stepwise voltage holding can be added to show the cathodic and anodic current to confirm the electrochemical stability.

Other minor issues:

Line 74, "worsen"

Line 242, "analyze"

Reviewer #2 (Remarks to the Author):

The authors studied quasi-solid electrolyte using MOF-based material with PC-based liquid electrolyte. Physical-chemical, electrochemical properties have been thoroughly investigated. The NMC coin cell and pouch cell using such quasi-solid electrolyte showed promising results. However, in general, the manuscript is lack of depth of investigation and understanding to explain the appealing results obtained. Hence, I would recommend to have a major revision by addressing the following questions before considering the publication.

- 1) quasi-solid electrolyte has stable electrochemical window upto 5.4V, author explained it as "The apparently enhanced voltage window of the quasi-solid electrolyte can be ascribed to the sub-nanoconfinement/coordination effects of MOF host towards the liquid electrolyte inside its channels and aggregative electrolyte configuration formed inside its sub-nano channels." This explanation describes the features of quasi-solid rather than the real reason for the high voltage window. Could author give more precise explanation?
- 2) The weight of quasi-solid electrolyte (3.5 mg cm^{-2}) is significantly lighter than LAGP (185.8 mg cm^{-2}) and PC liquid (33.9 mg cm^{-2}) electrolytes. Why 33.9 mg cm^{-2} is used in liquid cells? If use lower amount of liquid electrolyte in the cell, what is the performance?
- 3) There is no CEI on NCM cathode in quasi-solid cell, author should give a convincing explanation for such phenomenon.
- 4) CEI chemical components were probed by FTIR, why not XPS?
- 5) in page 11, "It should be noted that the SEM images of cycled NCM-811 cathodes from both typical liquid electrolyte (Figure S14a and S14b) and quasi-solid electrolyte (Figure S14c and S14d) demonstrated similar morphologies as shown in Figure 3a and 3b, Figure 3e and 3f, respectively." this conclusion is contradict to the results in the coin cell, where NCM cathode has CEI in liquid cell, while no CEI obtained in quasi-solid cell.
- 6) in page 11, "The extraordinary pouch-cell performances harvested under such harsh condition (being damaged and tested under 90 oC high temperature) suggested the significant importance of using the prepared quasi-solid electrolyte in constructing highly-safe and efficient LMBs." The conclusion should mention the reason leads to such significant importance.
- 7) Figure 1-3, too many texts and plots. The font size of number and texts in the plot are a bit small.

Title: “A stable quasi-solid electrolyte boosting highly-efficient lithium-metal pouch-cell work safely under harsh environment”

Manuscript number: NCOMMS-21-31759-T

Authors: Zhi Chang, Huijun Yang, Xingyu Zhu, Ping He and Haoshen Zhou*

We appreciate all of these valuable comments from the reviewers. Following are our responses to these comments from two reviewers.

Response to Reviewers:

To Reviewer #1:2-19

To Reviewer #2:20-32

Reviewer #1 (Remarks to the Author):

The manuscript by Chang et al. demonstrated a quasi-solid electrolyte by using a microporous metal-organic framework (MOF) with a small amount of liquid electrolyte. The integration of physical nanoconfinement effect and chemical coordination endows a strong interaction of MOF skeleton and electrolytes, which is desirable for the improvement of the high-temperature physicochemical properties. In addition, the nanopores of MOFs seems to also render highly ion-pairing solvation, which was found to be correlated to a stable cycling performance at both room and elevated temperatures. On the whole, the manuscript reads interesting and related characterization data have been collected to support their proposed concept. However, some of the descriptions on the experimental results as well as the discussion raise a number of questions that prevents the acceptance of this manuscript in its current state. I would suggest the suitable revisions to address the following concerns:

We are appreciating for your very positive comments and strong recommendation on this article. We are grateful for your support on this work. And we also thank you for your specific comments below. In general, we sincerely hope that the supplemented data added and the further explanations given can eliminate the reviewers' concerns.

1. It might be unreasonable to say an excellent conductivity in line 22, because the MOF trapped electrolyte shows noticeably reduced value compared with the bulk liquid as shown in Figure 3f.

Thank you very much for your constructive suggestion.

We are sorry for the inaccurate narrative we have made in our manuscript. In fact, we fully agree with the reviewer that after the MOF was used, the conductivity of the obtained quasi-solid electrolyte demonstrated noticeably decreased value when compared with the bulk liquid electrolyte. Even so, when compared with all-solid-state electrolyte, the quasi-solid electrolyte obtained in this work shown much superb conductivity. After kindly reminded by the reviewer, we have revised the corresponding inaccurate narrative and yellow highlighted in our revised manuscript.

“...Unlike typical liquid electrolyte under bulk state, in this work, we found that electrolyte confined in the sub-nanoscale environment (inside channels of 6.5 Å metal-organic framework, defined as quasi-solid electrolyte) exhibited eccentric properties and behaviors: higher boiling point; highly-aggregative configuration; decent lithium-ion conductivity; extended electrochemical voltage window (about 5.4 volts versus Li/Li⁺) and non-flammable under high working temperature...” (page 2, in the Abstract of revised manuscript)

“...Despite decent lithium-ion conductivity, the prepared quasi-solid electrolyte also demonstrated wide electrochemical stability window (about 5.4 volts versus Li/Li⁺)...” (page 4, in revised manuscript)

“...Moreover, due to the aggregative electrolyte configuration of the tiny liquid electrolyte

contained within the host material, the quasi-solid electrolyte exhibited decent lithium-ion conductivity and extended electrochemical voltage window (about 5.4 volts versus Li/Li⁺)...” (page 12, in revised manuscript)

We are sorry about this and sincerely looking forward to get support from the reviewer.

2. In line 103-107, the total mass of liquid plus MOF membrane and liquid plus the control separator should also be pointed out. Also, it is unfair to compare the amount of electrolyte in coin cells in line 105-107 to showcase the advantage of the quasi-solid electrolyte as the coin cell assembly cannot be scaled up directly to commercial cells with lean electrolyte.

Thank you very much for your valuable suggestion.

(1) We are sorry for not providing the total mass of quasi-solid electrolyte (liquid plus MOF) and liquid plus the control separator. We also fully agree with the reviewer that the total mass of both the prepared quasi-solid electrolyte and the control electrolyte should be supplemented since we would like to emphasize the superiority of our prepared quasi-solid electrolyte. After kindly reminded by the reviewer, we have supplemented the corresponding total mass loading of both the prepared quasi-solid electrolyte and the control electrolyte.

(2) For your second question, we also totally agree with the reviewer that we cannot directly compare the amount of electrolyte just in coin cells to emphasize the advantage of the prepared quasi-solid electrolyte. After kindly reminded by the reviewer, we also calculated the amount of electrolyte used in assembling pouch cell for a much fairer comparison. As previous works reported,^[R1, R2] for a practical pouch cell assembled with lean electrolyte, the electrolyte/capacity ratio of $\sim 3 \text{ g (Ah)}^{-1}$. Assuming the overall capacity of the pouch cell is 1 mAh cm^{-2} , the liquid electrolyte used was calculated to be 3.0 mg cm^{-2} . Considering weight of commercial separator employed (1.5 mg cm^{-2} as shown in Figure 3d in our original manuscript), the overall weight of separator & electrolyte used in fabricating pouch cell with lean electrolyte was calculated to be 4.5 mg cm^{-2} , which was still slightly higher than that of the quasi-solid electrolyte (3.5 mg cm^{-2} , as demonstrated in Figure 3d in our original manuscript) that prepared in this work. In other words, in this relatively fair comparison, the quasi-solid electrolyte reported in this work still has its advantage in weight. We have revised the corresponding section and yellow highlighted in our newly revised version.

“...Confined inside the sub-nano channels of flexible and porous metal-organic framework (6.5 \AA MOF), the quasi-solid electrolyte (total mass: $\sim 3.5 \text{ mg cm}^{-2}$) contained merely trace amount of liquid electrolyte ($< 0.23 \text{ \mu L cm}^{-2}$, equals to 0.3 mg cm^{-2}), which is apparently much lower than that of typical liquid electrolyte (total mass including separator: $\sim 34.0 \text{ mg cm}^{-2}$) used for LMBs assembling (about 25 \mu L cm^{-2} , equals to 32.6 mg cm^{-2}). The quasi-solid electrolyte also demonstrated its advantage in weight when compared with typical pouch cell assembled with lean electrolyte (3.5 mg cm^{-2} vs. 4.5 mg cm^{-2})...” (page 4, in revised manuscript)

[R1] J. Liu, Z. Bao, Y. Cui, E. J. Dufek, J. B. Goodenough, P. Khalifah, Q. Li, B. Y. Liaw, P. Liu, A. Manthiram, Nat. Energy 2019, 4, 180-186.

[R2] C. Niu, H. Lee, S. Chen, Q. Li, J. Du, W. Xu, J.-G. Zhang, M. S. Whittingham, J. Xiao, J. Liu, *Nat. Energy* 2019, 4, 551-559.

3. The sub-nanoconfinement effects are the basis of this manuscript and have been mentioned several times while no references or evidence to explain and support these concepts in the current liquid PC/MOF system. For the nanoconfinement of electrolytes for batteries, a recent paper has demonstrated similar phenomena in gaseous system (*Nat. Comm.*, 2021, 12, 3395). In addition, some comments are necessary to be added to highlight the difference in the concept as well as fundamentally address how the confinement occurs from molecular point of view.

Thank you very much for your constructive questions and suggestions.

We are sorry for not adding related references to explain and support our concepts in our original manuscript. We fully agree with the reviewer that for a rigorous scientific paper, we should make every sentence have a solid basis. This is our mistake for failing to add the corresponding citations in the original manuscript. After kindly reminded by the reviewer, we have added several important references including the *Nat. Communication* paper the reviewer recommended in the corresponding section and yellow highlighted in our revised manuscript. In addition, we also supplemented corresponding discussion and comments as the reviewer suggested in our revised manuscript to highlight the difference in the concept as well as fundamentally address how the confinement occurs from molecular point of view.

“...The remarkably improved decomposition temperature can be ascribed to the unique sub-nanoconfinement/coordination effect (physical confinement by narrow MOF channels and chemical interaction by metal sites inside channels) of the polar porous MOF host³²⁻³⁵ towards tiny amount of liquid electrolyte...” (page 6, in revised manuscript)

“...It worth noting that the sub-nanoconfinement/coordination effect reported in this work is different with those reported in other works. Since most of them reported sub-nanoconfinement/coordination effects towards aqueous solutions³²⁻³⁵, the sub-nanoconfinement/coordination effect demonstrated in this work focused on organic liquid electrolyte. Moreover, the most significant difference induced by the sub-nanoconfinement/coordination effect that reported in this work is the aggregative electrolyte configuration and the largely improved decomposition temperature of the tiny liquid electrolyte inside the narrow MOF channels (this will be discussed in detail in next section). We assumed the these unusual properties of the liquid electrolyte confined inside the MOF channels arise from the unique nature of sub-nano porous MOF material which combined large-scale flexibility with a heterogeneous polar internal surface (unsaturated polar Cu metal sides inside MOF channels enable strong physisorption or even chemisorption towards polar PC molecule solvents, can be verified by the (111) peak shown in Figure 2b) towards the polar liquid electrolyte molecule solvents used in this work...” (page 6, in revised manuscript)

“32. Otake, K.-i. et al. Confined water-mediated high proton conduction in hydrophobic channel of a synthetic nanotube. *Nature communications* 11, 1-7 (2020).

33. Rieth, A. J., Hunter, K. M., Dincă, M. & Paesani, F. Hydrogen bonding structure of confined water templated by a metal-organic framework with open metal sites. *Nature communications* 10, 1-7 (2019).

34. Ichii, T. et al. Observation of an exotic state of water in the hydrophilic nanospace of porous coordination polymers. *Communications Chemistry* 3, 1-6 (2020).

35. Cai, G. et al. Sub-nanometer confinement enables facile condensation of gas electrolyte for low-temperature batteries. *Nature communications* 12, 1-11 (2021).” (page 16, in revised manuscript)

We hope our explanation can satisfied the reviewer on this question and are also sincerely looking forward to get support from the reviewer.

4. It is hard to understand what is the coordination effect and how it works to endow a strong interaction between MOFs and electrolytes. Is it specific chemistry related? If PC is changed to other solvent, what will happen? In other words, the nanoconfinement should be a general phenomenon and needs to be demonstrated in a broader perspective.

Thank you very much for your valuable question.

This is a very good point. Actually, the question from the reviewer can be summarized in to two issues: (1) what is the coordination effect and the corresponding working mechanism; (2) universality of this effect.

(1) For your first question: we think the concerns/doubts were caused by our inadequate and unclear description. We are sorry for not giving enough discussion in detail in our original manuscript. And we think the concerns/doubts from the reviewer can be answered after we explain the structure of MOF in detail.

As schematically demonstrated in Figure R1a, for the pristine MOF, there are two types of solvents (water solvent in this stage) inside MOF channels: the sub-nano confined solvents (physically confined inside MOF channels) and the coordinated solvents (chemically coordinated by unsaturated Cu metal sites inside MOF channels). These two types of solvents were also reported by previous works^[R3, R4]. **(1-1)** Its XRD pattern was shown in Figure R1d. Clearly, there was apparent (111) peak (blue highlighted), which can be ascribed to the peak induced by coordinated solvents (water in this stage)^[R3]. However, as shown in Figure R1b and Figure R1e, after the MOF was experienced a typical activation process (heating the pristine MOF under vacuum at 200 °C for 12 hours to extrude any possible solvent inside MOF channels), the previous (111) peak just disappeared, which indicated the coordinated solvents were totally extruded after activation process^[R3]. As shown in Figure R1c and Figure R1f, after liquid electrolyte was introduced inside the MOF channels, the (111) peak appeared again, which suggested part of the liquid electrolyte was coordinated inside the MOF channels. So far, the existence of chemically coordinated solvents has been solidly confirmed by XRD patterns. **(1-2)** Yet, the XRD patterns fail to prove the existence of sub-nano confined solvents. To confirm the existence of sub-nano confined solvents, digital photos and TG curves of three different MOF samples were recorded as shown in Figure R1g-j. The sub-nano confined solvents were firstly confirmed by the different

colors of MOFs under different states as shown in Figure R1g. Obviously, for the pristine MOF (filled with both sub-nano confined solvents and chemically coordinated solvents), a light blue color can be clearly observed. For MOF activated under vacuum at 120 °C for 12 hours, a navy-blue color can be found, which can be ascribed to the loss of physically sub-nano confined solvents during the 120 °C vacuum activation. For MOF that activated under vacuum at 200 °C for 12 hours, a dark blue color can be clearly observed, which can be ascribed to the loss of both physically sub-nano confined solvents and chemically coordinated solvents^[R5]. The TG curve shown in Figure R1h indicated the loss of both physically sub-nano confined solvents and coordinated solvents while the TG curve shown in Figure R1i suggested the loss of chemically coordinated solvents.

Figure R1. Schematic illustration of sub-nano confined solvent and coordinated solvent inside MOF channels and the corresponding evidence to verify two different states of solvent inside MOF channels.

[R3] L. Shen, H. B. Wu, F. Liu, J. L. Brosmer, G. Shen, X. Wang, J. I. Zink, Q. Xiao, M. Cai, G. Wang, *Advanced Materials* 2018, 30, 1707476

[R4] H. K. Kim, W. S. Yun, M.-B. Kim, J. Y. Kim, Y.-S. Bae, J. Lee, N. C. Jeong, *Journal of the American Chemical Society* 2015, 137, 10009-10015

[R5] Z. Chang, Y. Qiao, H. Deng, H. Yang, P. He, H. Zhou, *Energy & Environmental Science* 2020, 13, 1197-1204

(2) For your second question: universality of this effect. Actually, this is a relatively macro question. Fortunately, there are some good news that we would like to share with you. In fact, based on our past research experiences and some of our recent findings, we are surprised to find that this effect is universal and can be achieved in different porous materials (with sub-nano channels/pore windows) coupled with different electrolyte systems. In general, we found not just limited to MOFs, porous materials like zeolites with proper pore windows (depends on the size of

solvent molecules of the target electrolyte) and excellent chemical/electrochemical stabilities (not only the prerequisites, but also the most important factors) are also considered as suitable candidates to achieve this effect. In other words, the strategy proposed in this work is universal. This strategy is expected to be work in other porous materials as long as one can carefully select proper porous materials and electrolytes. We have extended this finding from carbonate-based electrolyte into ether-based electrolyte and even aqueous electrolyte (our ongoing work) by using different porous materials.

(2-1) For example, another typical carbonate-based electrolyte (LiPF₆-EC/DMC electrolyte) by using a zeolite with small channels of 3.0 Å (0.3 nm) as shown in Figure R2. Clearly, FT-IR spectra in Figure R2b clearly indicated that electrolyte inside the zeolite channels was more aggregative than that of typical bulk diluent electrolyte (without using zeolite), and the electrolyte confined inside zeolite channels was mainly composed of strongly-coordinated solvents. The operando Raman also demonstrated the similar result (Figure R2c): after external electrical field was applied (charging the zeolite-based electrode front-face to different voltages), the strongly coordinated electrolyte solvents (strong Li⁺-EC interactions) became the dominated state while the former weakly-coordinated EC solvents can hardly be observed. The corresponding 3D color mapping picture shown in Figure R2d exhibited the same phenomenon. These spectroscopy results given directly evidence that the nanoconfinement effect reported in this work can also be achieved in carbonate-based electrolyte other than PC-based electrolyte.

Figure R2. Carbonate-based electrolyte confined inside the channels of 3.0 Å zeolite.

(2-2) We also found that this effect can also be achieved within ether-based electrolyte (LiTFSI-DME electrolyte) by using MOFs with different pore windows (9.0 Å (0.9 nm) and 6.5 Å (0.65 nm)) as shown in Figure R3. We found that the confinement effect was closely related to the pore size of the MOFs that employed. For example, as shown in Figure R3, for MOF with relatively large pore sizes (CuBTC, 9.0 Å), electrolyte confined inside CuBTC channels tends to formed a concentrated electrolyte. For MOF (PSS modified-CuBTC) with smaller pore size of about 6.5 Å, electrolyte confined inside channels formed a super-concentrated electrolyte configuration.

Figure R3. Ether electrolyte (LiTFSI-DME) confined inside channels of (a) CuBTC MOF (pore size: 9 Å, Concentrated electrolyte) and (b) PSS modified-CuBTC (pore size: 6.5 Å, Super-concentrated electrolyte).

(2-3) In addition to those organic electrolytes, we also found this effect can be found in aqueous electrolyte as shown in Figure R4. Obviously, when ZIF-7 MOF with 2.94 Å was introduced, the configuration of aqueous ZnSO₄ electrolyte inside the narrow ZIF-7 channels was even more aggregative (super-saturated electrolyte) than that of saturated electrolyte.

Figure R4. Aqueous electrolyte (ZnSO₄ aqueous electrolyte) confined inside channels of ZIF-7 MOF (2.94 Å window size).

These previous experimental results show that the nanoconfinement effect towards electrolytes is a general phenomenon and reproducible, and can be generalized to other electrolyte systems.

We sincerely hope our explanation can satisfied the reviewer on this question and are also sincerely looking forward to get support from the reviewer.

5. Why the quasi-solid electrolyte film has to be made by Li/Li symmetric cell and cycling for 10 cycles and the harvested from the cycled cells? How realistic is this method to be adopted in any practical cell fabrication application?

Thank you very much for your valuable question.

This is a very good point. The question from the reviewer can also be summarized in to two issues: (1) the reason for cycled within Li//Li symmetric cell for 10 cycles to prepared quasi-solid electrolyte and (2) possibility of practical application.

(1) For your first question: the reason we made the quasi-solid electrolyte by cycled it within Li//Li symmetric cell for 10 cycles is that we would like to employ the external electrical fields formed during cycling processes as the driving force to confine electrolyte inside MOF channels. Actually, due to their inherent sub-nano scale pore structures, microporous MOF was adopted as host materials to confine electrolyte. Yet, without additional driving forces, porous material alone cannot achieve the goal of confining electrolyte inside its channels. Fortunately, considering this process is expected to be carried out inside battery system, the external electrical fields can be used directly as the driving force to facilitate the electrolyte penetrated into MOF channels as schematically demonstrated in Figure R5. Under the effects of external electric field, weakly-coordinated solvents contained within the external solvation sheaths or even strongly-coordinated solvents within internal solvation sheaths are expected to be depleted and finally formed electrolyte with more aggregative electrolyte configuration confined inside MOF channels. The aggregative electrolyte configuration was also verified by the Raman result shown in Figure 3a and 3b in our original manuscript and Figure S3 in our original Supplementary Information.

Figure R5. Schematic illustration of the mechanism for preparing quasi-solid electrolyte in this work.

(2) For your second question, actually, to be honest, we also agree with the reviewer and admit that this method in fabricating electrolyte may bring some inconvenience in practical cell fabrication applications. Yet, we think this seemingly additional process can be effectively simplified in the future. (2-1) For example, in one of recently on-going work, we found that the electrochemical cycling time within Li//Li symmetric cell to harvest the quasi-solid electrolyte can be largely decreased to only 1 cycle. In this respect, this can greatly simplify the inconvenience in practical cell fabrication applications to some extent. (2-2) Another way that can simplify the process is inherit the technology that described in this article, making the MOF film as large as possible and assembled into a large pouch cell, then harvested after one cycling process. Finally, large amount of quasi-solid electrolyte can be obtained at one time. This method can also simplify the inconvenience in practical cell fabrication applications to some extent. However, we also admit that this kind of improvement is not enough in practical cell fabrication applications. To further simplify the fabrication of practical cell which using the quasi-solid electrolyte, other steps which can further simplify the process are also highly desired in the future. (2-3) Recently, we also tried to fabricate the quasi-solid electrolyte by using a new method combining improved

temperature (increase from under room temperature of 25 °C to 120 °C) and vacuumed pressure. Under this condition, liquid electrolyte can also be confined inside MOF channels. We think these methods mentioned can effectively simplify the whole process in the future.

We sincerely hope our explanation can satisfied the reviewer on this question and are also sincerely looking forward to get support from the reviewer.

6. Is any additional liquid electrolyte dropped in the cathodes? How the Li⁺ diffusion in the cathode? The experiment part describes the cell assembly by attaching a quasi-solid electrolyte to a cathode film, it could be possible to the cathode particles that in direct contact with the quasi-solid electrolyte layer to be redox active, but the particles inside the cathode that are far away from the electrolyte layer has no direct contact, how the ion transport occurs?

1) If pure solid-state diffusion, the kinetics will be super low for 20mg/cm². The authors should provide rate performance test show evaluate. Or 2) if the ion transport is supported by liquid, then the nanoconfinement effect may disappear/ be invalid through the cathode layer.

Or if there is any other way?

Thank you very much for valuable questions.

We must admit that this question from the reviewer is a good question, but it is also a difficult question to answer. But even so, we have tried our best to discuss the possible reasons of this problem in detail and hope our explanations can eliminate the concerns/doubts from the reviewer on this question. Firstly, for your first question, we would like to emphasize that during the battery assembling process, no any liquid electrolytes were added into the cathodes for the cell used the prepared quasi-solid electrolyte. We also agree with the reviewer that the wetting of cathode is important for battery to work normally. However, based on the excellent electrochemical performances of cells assembled with the prepared quasi-solid electrolyte, we think the quasi-solid electrolyte reported in this work can effectively solve the wetting issue concerns from the reviewer. We think this issue can be answered from two main directions: from (1) the mechanism level and (2) the technical engineering level.

(1) From the mechanism perspective: we think the tiny liquid electrolyte confined inside the MOF channels of the quasi-solid electrolyte is enough for wetting the cathodes.

(1-1) In fact, the majority of the conventional liquid electrolytes which added into typical batteries were consumed by the side-reactions occurred on the electrode/electrolyte interphases (cathode/electrolyte interphases and anode/electrolyte interphases) which finally led to the formation of thick CEI and SEI layers. Only small amounts of liquid electrolytes were served to wet the electrodes and acted as a medium for lithium ions' transportations.

(1-2) For the quasi-solid electrolyte prepared in this work, due to its unique electrolyte configuration (aggregative electrolyte configuration) and special physicochemical properties, we have obtained CEI-free cathode (as shown in Figure 4e-h in our original Manuscript) and dendritic-free lithium anode (as shown in Figure S12 in our original Supplementary Information). This suggested that the liquid electrolyte assumed by the side-reactions occurred on the electrode/electrolyte interphases were remarkably suppressed. This phenomenon is much similar to that of the pouch-cells which used lean electrolytes: pouch-cells under high cathode mass

loading (usually higher than 25 mg/cm^2) can also deliver excellent electrochemical performances when lean amounts of functional electrolytes (usually concentrated electrolytes) were used. Therefore, even only tiny amounts of liquid electrolyte were contained inside the MOF channels, we think it is still enough for wetting the cathodes.

(1-3) The XRD of the cycled quasi-solid electrolyte can also verify that the liquid electrolyte confined inside the MOF channels is enough for wetting the cathodes. As shown in Figure R6, even after cycling, the cycled (111) peak from the cycled quasi-solid electrolyte can still be clearly observed, which indicated there are still liquid electrolyte confined inside the MOF channels. Therefore, we think the tiny liquid electrolyte within the quasi-solid electrolyte is enough in wetting the cathodes.

(1-4) It will be more persuasive if liquid electrolyte can be found/detected inside the depth of cathode material layer. However, unfortunately, due to the limited experimental condition, in this work, the characterizations towards cathodes only concentrated on collecting the spectra information of the surface cathode material, it is still difficult to characterize whether there is liquid electrolyte inside the depth of the cathode materials. However, we will keep working on this field and try our best to find more accurate and persuasive reasons that lead to this in the near future.

Figure R6. XRD pattern of the cycled quasi-solid electrolyte.

(2) From the technical engineering perspective: Although we only attached the prepared quasi-solid electrolyte on the surface of the cathode material (instead of directly mixing the quasi-solid electrolyte particles and the cathode material and then coating them on the current collector which can greatly improve the contact between the cathode and quasi-solid electrolyte), considering the prepared quasi-solid electrolyte is flexible, we think the quasi-solid electrolyte after further mechanical extrusion on the cathode surface can be filled into the space gaps within the cathode material, thereby greatly improving the contact between the quasi-solid electrolyte and the particles inside the cathode.

(3) Benefiting from the greatly suppressed solvent-related decompositions on the cathode and the enhanced contact between cathode and the quasi-solid electrolyte induced by the flexibility of quasi-solid electrolyte, we at this stage assume that the tiny amounts of liquid electrolyte confined inside the MOF channels though seemly low, but can still effectively solving the wetting issue on the cathode. We also supplemented the rate performance of the NCM-811//Li cell assembled with the quasi-solid electrolyte as shown in Figure R7. Clearly, when assembled with the prepared quasi-solid electrolyte, excellent rate performance can also be achieved.

Figure R7. Rates performances of NCM-811//Li coin-cell used quasi-solid electrolyte.

We need to admit that all those characterizations used in this work were surface analyses on the electrode/MOF interfaces, but not the electrode/current collector interfaces. For typical liquid electrolyte battery systems, the electrolyte decomposition, SEI formation and accumulation are all happened on the interfaces/surfaces of electrodes/electrolytes. Therefore, we focused our research attention on this. The corresponding characterizations were also implemented on this part. However, to be honest, at current stage, it is highly difficult for us to characterize the configuration of electrolyte inside the deep depth of electrodes, in where only lean amount of electrolyte existed. However, various electrochemical results obtained in this work verify that lithium-ions can normally penetrate through the MOF layer and cathode layer.

In this study, we mainly focused on reporting a unique electrochemical/chemical phenomenon and gave primary evidences to support our finding. Deep exploration always requires more advanced deep-resolution characterizations to operando-test the information of electrolyte inside the deep depth of electrodes to further clarify the corresponding mechanism. So, it is currently out of now's content on this issue, to some extent. But we will keep research on this topic and try to find out more valuable information in the future.

We sincerely hope our explanation can satisfied the reviewer on this question and are also sincerely looking forward to get support from the reviewer.

7. Any comments on the selection of MOFs and polymers for the construction of MOF-based porous hosts? Any comment on the polytetrafluoroethylene (PTFE) as the polymer binder? In addition, what are the purposes/functions of the polytetrafluoroethylene (PSS) in this work?

Thank you very much for your valuable question.

(1) For your first question about the critierion in selecting MOFs and polymers, actually this is a relatively macro problem. (1-1) In the selection of MOF materials, we believe that the desirable MOFs need to meet the following characteristics: easy to synthesized, possess appropriate pore window sizes (depending on the type of electrolytes), chemically stable, electrochemically stable, and compatible towards both cathodes and anodes. As for the selection of binders, actually, we think ideal binder needs to have excellent bonding properties, good stability (not only chemically, but electrochemically) and flexibility. In addition, the use of binder to prepare MOF film should be not only convenient and time-saving, but environmentally friendly as possible. (1-2) For

example, in our work, we used PTFE binder because in the process of preparing MOF film, water will inevitably be introduced, so we choose water-stable PTFE as the binder. At the same time, using PTFE to prepare MOF film is not only convenient and time-consuming, but also environmental friendliness (since environmental harmful NMP solvents was totally avoided). Moreover, considering the subsequent physical extrusion process implemented to the prepared MOF film, MOF film prepared by using PTFE possessed very good toughness (not brittle). This makes it an ideal binder compared with other binders.

(2) For your second question about the purposes of introducing polytetrafluoroethylene (PSS) in this work, actually, we think there are two main reasons: (2-1) decreases the channel sizes of MOF and (2-2) facilitates the transportations of lithium-ions. As shown in the (2-2) part in our reply to your 4th comment, we found the electrolyte configuration of the electrolyte confined inside MOF channels are closely related to the pore sizes of MOF. In order to promote electrolyte inside MOF channels into more aggregative state (as demonstrated in Figure 3a and 3b), we decorated the channels of CuBTC MOF (9.0 Å pore size) with PSS to decrease the MOF channel to slightly narrower one (6.5 Å). On the other hand, the PSS decorated inside MOF channels was negatively charged. The negative charged PSS groups the MOF channels an electronegative environment, which can consequently facilitate the transport kinetics of lithium ions thus exempt MOF away from cumbersome and awkward lithium ions transport energy barrier^[R6, R7, R8].

We sincerely hope our explanation can make it clear for the reviewer on this question and are also sincerely looking forward to get support from the reviewer.

[R6] Y. Guo, Y. Ying, Y. Mao, X. Peng, B. Chen, *Angew. Chem. Int. Ed.* 2016, 128, 15344-15348.

[R7] Z. Chang, Y. Qiao, J. Wang, H. Deng, P. He, H. Zhou, *Energy Storage Mater.* 2020, 25, 164-171.

[R8] Q. Zeng, J. Wang, X. Li, Y. Ouyang, W. He, D. Li, S. Guo, Y. Xiao, H. Deng, W. Gong, *ACS Energy Lett.* 2021, 6, 2434-2441.

8. Any comment on the long-term effects or on the projected shelf life of the batteries with the quasi-solid electrolyte?

Thank you very much for your valuable question.

This is a very good point. We fully agree with the reviewer that the long-term stability of batteries assembled with the quasi-solid electrolyte is important. Actually, in our original manuscript and Supplemental Information, the long-term stability of batteries assembled with the quasi-solid electrolyte have been verified by the corresponding good cycling performances (Li//Li symmetric cell; LFP//Li, LTO//Li and NCM-811//Li coin-cell; NCM-811//Li pouch-cell).

In fact, even during the manuscript submission period, some batteries are still running. As one of the most representative battery, we have updated the electrochemical performance of the NCM-811//Li coin cell and refreshed the corresponding Figure S11a in our revised Supplementary Information as shown in Figure R8. Clearly, NCM-811//Li coin-cell used quasi-solid electrolyte (under room-temperature of 25 °C) exhibited good cycling stability even after 700 cycles. In addition, compared with the limited cycling life and fast capacity decay of NCM-811//Li pouch

cells used typical liquid electrolyte (Figure 5d and 5e), the largely improved cycling life and high capacity of NCM-811//Li pouch cells assembled with quasi-solid electrolyte can to some extent, prove the important role of quasi-solid electrolyte in promoting the long-term life of the batteries.

Figure R8 (now as refreshed Figure S11, extended from 500 cycles to 700 cycles). (a) cycling performance and the corresponding (b) discharge/charge curves of the NCM-811 coin-cell (NCM-811 cathode mass loading of about 20 mg cm⁻²) used quasi-solid electrolyte under room-temperature (25 °C).

Stable cathode and anode are also important parameter for batteries to achieve long-term stability. Apart from the largely improved electrochemical performances of batteries assembled with quasi-solid electrolyte, the dendritic-free Li anode (Figure S12) and smooth NCM-811 cathode (without too many undesirable CEI, Figure 4e and 4f) together also exhibited the promising potential of the quasi-solid electrolyte in promoting the long-term life of the batteries.

In addition, not limited to electrochemical cycling stability of batteries and stability of cycled anodes and cathodes, the stability of the quasi-solid electrolyte itself is also an important factor in determining whether batteries (assembled with quasi-solid electrolyte) can be cycled stably for long life. To further study the stability of the quasi-solid electrolyte, we also measured the XRD pattern and SEM image of the cycled quasi-solid electrolyte (Figure R9). Obviously, as shown in Figure R9a and 9b, even after long-term cycling, the cycled quasi-solid electrolyte still maintained its compact structure which composed of numerous closely attached MOF particles. The XRD pattern in Figure R9b demonstrated that the cycled quasi-solid electrolyte did not experience obvious physical degradation after cycling. These results indicated the excellent stability of the quasi-solid electrolyte even after various electrochemical cycling processes.

Figure R9 (now as Figure S15 in our revised Supplementary Information). Stability of the cycled quasi-solid electrolyte. (a, b) SEM images of the cycled quasi-solid electrolyte. (c) XRD pattern of the cycled quasi-solid electrolyte.

The XRD pattern and SEM image of the cycled quasi-solid electrolyte and supplemented in our revised Supplementary Information as Figure S15. The corresponding discussion was also added and yellow highlighted in our revised manuscript.

“...Moreover, the SEM images and the corresponding XRD result shown in Figure S15 suggested the excellent stability of the quasi-solid electrolyte even after various electrochemical cycling processes. These obtained results together exhibited the promising potential of the quasi-solid electrolyte in promoting the long-term life of the batteries...” (page 11, in revised manuscript)

We sincerely hope our explanation can satisfied the reviewer on this question and are also sincerely looking forward to get support from the reviewer.

9. What is the scientific value of fig 1? It’s more like something designed for a review paper.

Thank you very much for your valuable question.

In fact, all figures, whether those are in research articles or in review papers, are generally used to serve the central idea of the article. The first picture of the article is often particularly important, because it will clearly show the key points and core ideas and significance of an article at a glance.

In this article, our core idea is to highlight the importance of designing and using quasi-solid electrolytes in rechargeable lithium-metal batteries. However, we thought directly putting the quasi-solid electrolyte on it at the beginning will appear a bit abrupt. Therefore, we decided to start from analyzing the current wide applications of lithium-ion batteries, their inspiring prospects and series of serious problems existing in lithium-ion batteries that assembled with typical liquid electrolytes. Through a simple summary, we thought it will be much easier for potential readers to

quickly understand the importance of preparing and using quasi-solid electrolytes. This is the reason why we designed the Figure 1 in that form in our original manuscript.

However, after tasting your comment repeatedly, we begun to realize that some of the pictures in Figure 1 in our original manuscript need to be re-designed. Also, as we have discussed in the last paragraph, in our original Figure 1, we only talked about **why** we need to focus our attention on preparing quasi-solid electrolyte, but **failed to answer the question of how** to achieve it. Inspired by the reviewer, we decided to re-design our Figure 1 as shown in Figure R10. By adding the Figure R10c, we thought the question of **how** to achieve the quasi-solid electrolyte

Figure R10 (now as Figure 1 in our revised manuscript). The importance of using non-liquid electrolyte in constructing safe lithium-metal batteries. (a) Safety hazards which may induced by using traditional liquid electrolytes. (b) Radar chart of the advantages of quasi-solid electrolytes (compared with typical liquid electrolyte and solid-state electrolyte). (c) Lithium-metal battery assembled with proposed quasi-solid electrolyte.

The Figure R10 was hence employed as our newly Figure 1 in our revised manuscript. In addition, due to the changes in Figure 1, we also revised the corresponding sentences in Introduction part and yellow highlighted in our newly revised manuscript.

“...Batteries, especially the lithium-ion batteries (LIBs)^{3, 4}, are widely used to power various electronic devices (from portable electric devices like hand watches to smart phones, laptops and even electric vehicles (EVs)). The rapid development and ever-increasing use of electronic equipment puts great demands on not only their densities and cycling lives, but the productions and shipments of LIBs. According to the report, both the global LIBs market and LIBs’s shipments have gradually increased during the past three years, and are also expected to steadily climbing in the coming five years⁵. Specifically, the global LIBs market and shipment in 2025 are predicted to be 108.9 billion dollars, 439.3 GWh, respectively⁵...” (page 3, in revised manuscript)

“...LMBs, especially the high-voltage LMBs which assembled with typical liquid electrolytes tend to suffer serious electrolyte degradations which were induced by the high reactivity between liquid electrolytes and charged transition metal oxide surfaces/reactive lithium-

metals (Figure 1a)^{10,20}...” (page 3, in revised manuscript)

“...As an intermediate state between liquid electrolyte and solid electrolyte, quasi-solid electrolyte would take the advantages of liquid electrolyte and solid electrolyte while avoiding the shortcomings of both sides (Figure 1b)^{24, 27}. Quasi-solid electrolyte, can not only provide mechanical stiffness to block dendrites, but deliver much safer operation environment (non-flammable) compared to typical liquid electrolyte. On the other hand, quasi-solid electrolyte also possesses higher ion conductivity and superior interfacial property than that of solid-state electrolyte. Confining lean amounts of liquid electrolyte inside the host matrix that possesses nano porous (sub-nano porous) structure as shown in Figure 1c is a promising method to prepare a quasi-solid electrolyte which can meet the above requirements discussed in Figure 1b (good contact towards electrodes, hard to be volatilized, stable and safe under high working temperature)...” (page 4, in revised manuscript)

We sincerely appreciate the constructive suggestion from the reviewer for helping our work more logical and complete.

We also sincerely hope our explanation can satisfied the reviewer on this question and are sincerely looking forward to get support from the reviewer.

10. It is surprising that Cu-based MOF can be stable in such wide voltage window. CA experiment with stepwise voltage holding can be added to show the cathodic and anodic current to confirm the electrochemical stability.

Thank you very much for your constructive suggestion.

(1) We think the Cu-based MOF it self is stable. (1-1) If the Cu metal inside the MOF channels were oxidized during various electrochemical performance, then the coordination structure of the MOF material will be destroyed, which would lead to a totally different XRD pattern. To study the stability of the Cu-based MOF used in this work, we also measured the XRD pattern and SEM image of the cycled quasi-solid electrolyte (Figure R9). Obviously, as shown in Figure R9a and 9b, even after long-term cycling, the cycled quasi-solid electrolyte still maintained its compact structure which composed of numerous closely attached MOF particles. The XRD pattern in Figure R9b demonstrated that the cycled quasi-solid electrolyte did not experience obvious physical degradation after cycling. These results indicated the excellent stability of the Cu-based MOF even after various electrochemical cycling processes (including under wide voltage window). (1-2) To further study the stability of the Cu-based MOF used in this work, we also measured the Cu 2p XPS spectra of the pristine and cycled Cu-MOF based quasi-solid electrolyte that used in this work. (Figure R11). Clealry, the cycled Cu-MOF based quasi-solid electrolyte did not exhibit any difference compared with its pristine counter-part. The XPS result demonstrated in Figure R11 again indicated the Cu-based MOF is stable.

Figure R9 (now as Figure S15 in our revised Supplementary Information). Stability of the cycled quasi-solid electrolyte. (a, b) SEM images of the cycled quasi-solid electrolyte. (c) XRD pattern of the cycled quasi-solid electrolyte.

Figure R11. Cu 2p XPS spectra of the pristine and cycled Cu-MOF used in this work.

(2) We think the Cu-MOF based quasi-solid electrolyte is also stable. We also fully agree with the reviewer that the electrochemical stability of the MOF-based quasi-solid electrolyte is important. Although we have already provided the LSV curve of the quasi-solid electrolyte in our original manuscript, we think this is not enough. With the kind suggestions of the reviewer, we have supplemented some additional data to further verify the electrochemical stability of the prepared MOF-based quasi-solid electrolyte. Potentiostatic Intermittent Titration Technique (PITT) of both liquid electrolyte (1M LiTFSI-PC electrolyte) and quasi-solid electrolyte were measured. As shown in Figure R12a, during the voltage floating process, cell tested in typical 1M LiTFSI-PC electrolyte demonstrated obvious floating currents (black curves in Figure R12a). For sharp contrast, cell tested in quasi-solid electrolyte demonstrated apparently much lower floating currents (blue curves in Figure R12a) than that of typical electrolyte. Results shown in Figure R12a verified the superior electrochemical oxidative stability of the prepared quasi-solid electrolyte under high voltage. In addition, cyclic voltammograms (CV) curves of two electrolytes between 0 and 3.0 V were also tested. Clearly, as shown in Figure R12b, the cathodic sweeping curve of liquid electrolyte (1M LiTFSI-PC electrolyte) during the initial cycle (Figure R12b, black curves) exhibited large cathodic current, indicated serious electrolyte decomposition. More

seriously, the decomposed products would be re-oxidized during the following anodic sweeping. Obvious electrolyte cathodic current and anodic current can still be clearly observed during the second sweeping, suggested poor surface layer formation ability of typical 1M LiTFSI-PC electrolyte. For stark contrast, the quasi-solid electrolyte demonstrated both greatly reduced cathodic current and anodic current, which suggested largely enhanced surface layer formation ability (Figure R12b, blue curves). The newly added results shown in Figure R12 combined with the LSV result demonstrated in Figure 3c further verified the excellent electrochemical stability of the prepared quasi-solid electrolyte.

Figure R12 (now as Figure S3g and 3h in our revised Supplementary Information). (a) Potentiostatic Intermittent Titration Technique floating test and (b) Cyclic voltammograms of typical liquid electrolyte (1M LiTFSI-PC electrolyte) and the prepared quasi-solid electrolyte.

We sincerely appreciate your highly constructive suggestion and supplemented the data shown in Figure R12 into Figure S3 in our revised Supplementary Information. In addition, we also added the corresponding discussion and yellow highlighted in our revised manuscript.

“...Obviously, the quasi-solid electrolyte exhibited apparent extended electrochemical stability window to 5.4 V (blue curve in Figure 3c), which was obvious much higher than that of typical liquid electrolytes (green curve in Figure 3c). The Potentiostatic Intermittent Titration Technique (PITT) results in Figure S3g and cyclic voltammograms (CV) curves in Figure S3h further verified the excellent electrochemical stability of the prepared quasi-solid electrolyte. The excellent electrochemical stability of the quasi-solid electrolyte can be ascribed to the sub-nanoconfinement/coordination effects of MOF host towards the liquid electrolyte inside its channels and aggregative electrolyte configuration formed inside its sub-nano channels...” (page 6, in revised manuscript)

Figure S3. Raman and ATR-FTIR spectra of (a, b) PC solvent, (c, d) 1M typical liquid and (e, f) concentrated electrolyte. (g) Potentiostatic Intermittent Titration Technique floating test and (h) Cyclic voltammograms of typical liquid electrolyte and the prepared quasi-solid electrolyte.

We sincerely appreciate your highly constructive suggestion and also sincerely looking forward to get support from the reviewer.

Other minor issues:

Line 74, “worsen”

Line 242, “analyze”

Thank you very much for your constructive suggestion.

We are terribly sorry about all those spelling, grammar errors and other mistakes we have made in our original manuscript. Moreover, we have checked our manuscript very carefully in order to avoid any unclear narrative, possible errors and typos. And the corresponding changes were also yellow highlighted in our revised manuscript.

“...Even worse, various gaseous products produced during charge/discharge processes and the potential shortcut after the dendritic lithium pierced the separator would potentially lead to serious safety hazards like battery burning or even horrible battery explosion especially when batteries are working under high temperature²²...” (page 4, in revised manuscript)

“...It should be noted that these two characteristic peaks can be detected almost during the whole argon etching process, which gave direct quantitative analysis towards the thickness (50 nm in total) of CEI layer...” (page 8, line 242, in revised manuscript)

Reviewer #2 (Remarks to the Author):

The authors studied quasi-solid electrolyte using MOF-based material with PC-based liquid electrolyte. Physical-chemical, electrochemical properties have been thoroughly investigated. The NMC coin cell and pouch cell using such quasi-solid electrolyte showed promising results. However, in general, the manuscript is lack of depth of investigation and understanding to explain the appealing results obtained. Hence, I would recommend to have a major revision by addressing the following questions before considering the publication.

We are appreciating for your very positive comments and strong recommendation on this article. We are grateful for your support on this work. And we also thank you for your specific comments below. In general, we sincerely hope that the supplemented data added and the further explanations given will eliminate your concerns.

(1) quasi-solid electrolyte has stable electrochemical window up to 5.4V, author explained it as “The apparently enhanced voltage window of the quasi-solid electrolyte can be ascribed to the sub-nanoconfinement/coordination effects of MOF host towards the liquid electrolyte inside its channels and aggregative electrolyte configuration formed inside its sub-nano channels.” This explanation describes the features of quasi-solid rather than the real reason for the high voltage window. Could author give more precise explanation?

Thank you very much for your valuable question and constructive suggestion.

We apologize for failing to provide a deeper explanation. And after kindly reminded by the reviewer, we summarized to main reasons for the high voltage window of the prepared quasi-solid electrolyte as shown below.

In fact, the behaviors of the quasi-solid electrolyte were closely related to its electrolyte configuration and the environment it stayed in. We think the high voltage window of the quasi-solid electrolyte can be ascribed to the aggregative electrolyte configuration induced by the sub-nano confinement environment.

Compared with typical diluent liquid electrolyte (1M LiTFSI-PC, Figure S3c and S3d), tiny liquid electrolyte confined inside MOF channels demonstrated more stronger Li-PC interaction and concentrated TFSI⁻ state, suggested much aggregative electrolyte configuration. It is worth noting that tiny liquid electrolyte confined inside MOF channels was even more aggregative than concentrated liquid electrolyte (3M LiTFSI-PC, Figure S3e and S3f). Due to the enhanced Li-PC solvents and Li-TFSI⁻ interactions, the solvation sheaths of lithium-ions became smaller but much compact. Therefore, it was more difficult for the solvated PC solvents to be removed from the solvation sheaths of the solvated lithium ions. Compared with dilute electrolyte, in order to oxidize and decompose the quasi-solid electrolyte, additional power is needed: a higher voltage when reflected in electrochemical processes. This is also the reason why concentrated electrolytes possessed much higher electrochemical stability windows than their diluent electrolyte counterparts^[R9, R10, R11].

We think the confinement effect created by the sub-nano channels towards the tiny liquid electrolyte inside the MOF channels also contributed to the high electrochemical voltage window of the quasi-solid electrolyte. As the result shown in Figure 2e and 2f in our original manuscript,

due to the sub-nano confinement effect created by the sub-nano channels of MOF, tiny liquid electrolyte confined inside the MOF channels tend to possess much higher decomposition temperature (compared with typical liquid electrolyte). Compared with typical liquid electrolyte, in order to oxidize and decompose the quasi-solid electrolyte, additional force is needed: a higher voltage when reflected in electrochemical processes.

In summary, we think both the aggregative electrolyte configuration and the sub-nano confinement environment created by the MOF channels contributed to the high electrochemical voltage window of the quasi-solid electrolyte. The corresponding discussion was also revised and yellow highlighted in our revised manuscript.

“...The apparently enhanced voltage window of the quasi-solid electrolyte can be ascribed to the sub-nanoconfinement/coordination effects of MOF host towards the liquid electrolyte inside its channels and aggregative electrolyte configuration formed inside its sub-nano channels. To be more specific, on the one hand, more aggregative electrolyte configuration usually means enhanced Li-PC solvents and Li-TFSI⁻ interactions. Due to the smaller but much compact solvation sheaths of lithium-ions, it was more difficult for the solvated PC solvents to be removed from the solvation sheaths of the solvated lithium ions and then be oxidated. Thus, compared with dilute electrolyte, the quasi-solid electrolyte exhibited apparently enhanced voltage window. On the other hand, compared with typical liquid electrolyte, due to the sub-nano confinement effect created by the sub-nano channels of MOF, tiny liquid electrolyte confined inside the MOF channels tend to possess much higher decomposition temperature (as shown in Figure 2e and 2f). Compared with typical liquid electrolyte, in order to oxidize and decompose the quasi-solid electrolyte, additional force is needed. This is the other reason for high voltage window of the quasi-solid electrolyte...” (page 6, in revised manuscript)

We sincerely appreciate your highly constructive suggestion and also sincerely looking forward to get support from the reviewer.

(2) The weight of quasi-solid electrolyte (3.5 mg cm^{-2}) is significantly lighter than LAGP (185.8 mg cm^{-2}) and PC liquid (33.9 mg cm^{-2}) electrolytes. Why 33.9 mg cm^{-2} is used in liquid cells? If use lower amount of liquid electrolyte in the cell, what is the performance?

Thank you very much for your valuable question.

Actually, according to our past experimental experience and other results from other works, under normal circumstances when coin cells were assembled, about 50 μL of liquid electrolyte need to be added to every coin cell. In this work, liquid 1M LiTFSI-PC electrolyte was used as the controlled electrolyte for comparisons. 50 μL liquid 1M LiTFSI-PC electrolyte equals to 67 mg cm^{-2} in weight (68.5 mg cm^{-2} including the weight of separator). However, in order to compare the weight difference between the quasi-solid electrolyte and liquid 1M LiTFSI-PC electrolyte in a more fairly way, the amount of the liquid 1M LiTFSI-PC electrolyte we adopted was only 25 μL (equals to 32.6 mg cm^{-2} , and 33.9 mg cm^{-2} considering the weight of separator), which is already a relatively low electrolyte value during coin cell fabrications.

After kindly suggested by the reviewer, we have supplemented the electrochemical

performance of NCM-811//Li coin cell assembled with even lower electrolyte amount of 13 μL (16.9 mg cm^{-2}). The electrochemical performance of NCM-811//Li coin cell assembled with electrolyte amount of 25 μL was also measured as comparison. Obviously, as shown in Figure R13, NCM-811//Li coin cell assembled with 13 μL liquid electrolyte demonstrated much fast capacity decay than that of the 25 μL liquid electrolyte used cell. It worth noting that the cycling performances of two NCM-811//Li coin cells assembled with low amounts of liquid electrolytes were much worse than the electrochemical performance of the NCM-811//Li coin cell assembled with the quasi-solid electrolyte as shown in Figure S11 in our original Supplementary Information.

Figure R13. Cycling performances of NCM-811//Li coin cells assembled with low liquid electrolyte amount of 25 μL and 13 μL .

We also sincerely hope our explanation can satisfied the reviewer on this question and are sincerely looking forward to get support from the reviewer.

(3) There is no CEI on NCM cathode in quasi-solid cell, author should give a convincing explanation for such phenomenon.

Thank you very much for your valuable suggestion.

We are pretty sorry for not given detail explanation on the phenomenon of nearly no CEI can be observed on cycled NCM cathode.

We think this interesting phenomenon can be ascribed to the special physiochemical properties/behaviors of the prepared quasi-solid electrolyte. Similar to our reply in your first comment, we think the behaviors of the quasi-solid electrolyte were closely related to its electrolyte configuration and the environment it stayed in, that is: the aggregative electrolyte configuration induced by the sub-nano confinement environment.

Compared with typical diluent liquid electrolyte (Figure S3c and S3d), tiny liquid electrolyte confined inside MOF channels demonstrated more stronger Li-PC interaction and concentrated TFSI⁻ state, suggested more aggregative electrolyte configuration. It is worth noting that tiny liquid electrolyte confined inside MOF channels was even more aggregative than concentrated liquid electrolyte (3M LiTFSI-PC, Figure S3e and S3f). Due to the enhanced Li-PC solvents and Li-TFSI⁻ interactions, the solvation sheaths of lithium-ions became smaller but much compact. Correspondingly, the number of PC solvent molecules contained within each solvation sheath of the solvated lithium-ion (tiny liquid electrolyte confined inside MOF channels, contained only strongly coordinated PC solvents as shown in Figure 3a in our original manuscript) are greatly reduced compared to that of both the dilute electrolyte (which contained various free PC solvents

and weakly-coordinated PC solvents within the solvated lithium-ions as exhibited in Figure S3c in our original Supplementary Information) and the concentrated electrolyte (contained both various weakly-coordinated and strongly-coordinated PC solvents within the solvation sheaths of solvated lithium-ions as exhibited in Figure S3e in our original Supplementary Information).

Generally, for most secondary batteries, the solvated metal-ions tend to experience typical de-solvation process during the discharge/charge procedures. The typical de-solvation processes of solvated metal-ions, which generally occurred on the electrode/electrolyte interphases, played important role in influencing the components, morphologies and ionic conductivities of solid electrolyte interlayers/cathode electrolyte interlayers (SEIs/CEIs). This classic process resulted various detrimental issues existed in both the cathode/electrolyte interphases and anode/electrolyte interphases, and finally significantly affected the mutual matching of electrodes and electrolytes, as well as the structure of the secondary batteries. The de-solvation processes of solvated metal-ions contributed to the formation of de-solvated electrolyte solvents, which was one of the three types of solvent molecules confronted on electrode surface. The other two kinds of were free solvents and sheath solvents (including weakly-coordinated solvents and strongly-coordinated solvents), respectively. The de-solvated solvents were the final state of sheath solvents (as shown in Figure R14). Those existed solvent molecules would bring various detrimental solvent decomposition issues to both cathodes and anodes. On the cathode side, those existed solvent molecules can be easily oxidized when operated under high working voltage (e.g., nonaqueous electrolytes: dehydrogenation; aqueous electrolytes: oxygen evolution reaction (OER)), and lead to limited electrolyte electrochemical voltage windows. More seriously, cathode materials, especially the high-voltage cathodes, which usually contained highly catalytic transition metal oxides (e.g., high-Ni $\text{LiNi}_{0.8}\text{Co}_{0.1}\text{Mn}_{0.1}\text{O}_2$ (NCM-811); $\text{LiNi}_{0.5}\text{Mn}_{1.5}\text{O}_4$ (LNMO)) would attack the solvent molecules and accelerate the decomposition of electrolytes when under charged states. This would consequently lead to the formation of undesirable cathode/electrolyte interphases, and finally deteriorated the electrochemical performances of batteries.

Figure R14. The significance of tailoring the solvation sheath of electrolyte by constructing electrode front-face for rechargeable battery.

In this work, benefits from the eliminated free PC solvents and weakly-coordinated PC solvents as well as part of the strongly-coordinated PC solvents within solvated lithium-ions of the quasi-solid electrolyte, by-products related to the decomposition of PC solvent molecules are greatly reduced. Therefore, we think this interesting phenomenon can be ascribed to the special physiochemical properties/behaviors of the prepared quasi-solid electrolyte. The corresponding discussion was also added and yellow highlighted in our revised manuscript.

“...This means after the quasi-solid electrolyte was used, NCM-811 cathode experienced significantly reduced electrolyte decomposition and greatly improved electrolyte/cathode

interphase (CEI-free cathode surface). We thought this can be ascribed to the aggregative electrolyte configuration of the quasi-solid electrolyte induced by the sub-nano confinement effect of MOF channels. Due to the enhanced Li-PC solvents and Li-TFSI⁻ interactions, the solvation sheaths of lithium-ions became smaller but much compact. And the number of PC solvent molecules contained within each solvation sheath of the solvated lithium-ion (tiny liquid electrolyte confined inside MOF channels, contained only strongly coordinated PC solvents as shown in Figure 3a in our original manuscript) are greatly reduced compared to that of both the dilute electrolyte (which contained various free PC solvents and weakly-coordinated PC solvents within the solvated lithium-ions as exhibited in Figure S3c) and the concentrated electrolyte (contained both various weakly-coordinated and strongly-coordinated PC solvents within the solvation sheaths of solvated lithium-ions as exhibited in Figure S3e). Benefited from the eliminated free PC solvents and weakly-coordinated PC solvents as well as part of the strongly-coordinated PC solvents within solvated lithium-ions of the quasi-solid electrolyte, by-products related to the decomposition of PC solvent molecules are greatly reduced, thus lead to a nearly CEI-free NCM-811 cathode...” (page 8, in revised manuscript)

We also sincerely hope our explanation can satisfied the reviewer on this question and are sincerely looking forward to get support from the reviewer.

(4) CEI chemical components were probed by FTIR, why not XPS?

Thank you very much for your valuable suggestion.

In fact, the functions of the Etching FTIR and Etching XPS (depth-resolution) are similar in general: both of them can provide accurate information about the CEI thickness and the components of the CEI layers that covered on the surface of cycled NCM-811 cathodes. However, the FTIR and XPS also has slight differences when employed to characterized the CEI components of the cycled cathodes. Generally, FTIR is capable of providing accurate information of organic products very easily, but fail to detect accurate/useful information about inorganic products, while XPS is more accurate in detecting inorganic products.

After kindly reminded by the reviewer, we also added the data of Etching XPS towards the cycled NCM-811 cathodes and hoped to make up for the shortcomings of Etching FTIR. As shown in Figure R15a (O 1s) (now as Figure S7c in our revised Supplementary Information), despite the lattice oxygen (529.5 eV), the NCM-811 cathode harvested from the NCM-811//Li cell assembled with typical liquid electrolyte (1M LiTFSI-PC electrolyte) demonstrated various PC solvents decomposition related by-products like carboxylates, carbonates. PVDF can also be detected among the whole etching process. While for sharp contrast, the NCM-811 cathode harvested from the quasi-solid electrolyte used NCM-811//Li cell did not exhibit any PC solvent induced byproducts. In addition, as shown in Figure R15b (F 1s) (now as Figure S7d in our revised Supplemental Information), despite the PVDF related peaks, the NCM-811 cathode harvested from the NCM-811//Li cell assembled with typical liquid electrolyte demonstrated only tiny LiF, while thick LiF can be detected on the cycled NCM-811 cathode that harvested from the quasi-solid electrolyte used NCM-811//Li cell. Those characterizations towards cycled NCM-811 cathodes suggested that the quasi-solid electrolyte can enabled significantly suppressed PC solvent

decomposition on the surface of NCM-811 cathode.

Figure R15. The corresponding (a) O1s and (b) F1s XPS results of the cycled NCM-811 cathodes.

By combining the two, we can get accurate information about both the CEI thickness and the components of the CEI layer. We sincerely appreciate your highly constructive suggestion and supplemented the data shown in Figure R15 into Figure S7 in our revised Supplementary Information. The corresponding discussion was also supplemented and yellow highlighted in our revised manuscript. The experimental section was also revised by adding the etching XPS measurement and yellow highlighted in our Supplemental Information.

Figure S7. Selected etching FT-IR spectra recorded from the cycled NCM-811 cathode cycled within (a) quasi-solid electrolyte and (b) typical liquid electrolyte. (c) O 1s and (d) F 1s of selected etching XPS spectra recorded from cycled NMC-811 cycled within quasi-solid electrolyte and typical liquid electrolyte. (top: NMC-811 typical liquid electrolyte; bottom: NMC-811 cycled within quasi-solid electrolyte).

“...The corresponding results from the depth-resolution etching XPS (Figure S7c and 7d) consists well with etching FT-IR results shown in Figure 4c. This means after the quasi-solid electrolyte was used, NCM-811 cathode experienced significantly reduced electrolyte decomposition and greatly improved electrolyte/cathode interphase (CEI-free cathode surface)...” (page 8, in revised manuscript)

“...**Depth-resolution Etching X-ray Photoelectron Spectroscopy (XPS) Characterization** XPS measurement was performed using a VG scientific ESCALAB 250 spectrometers with monochromic Al K α Ka source (1486.6 eV) under ultra-high vacuum. Similarly, to prevent long-time exposure to air environment, the samples (after rinsing and dry) were tightly sealed into an Ar-filled bottle and then soon transferred into XPS chamber as quickly as possible. The XPS was equipped with etching with different depth to analysis the component distribution...” (page 7, in revised Supplemental Information)

We sincerely appreciate your highly constructive suggestion and also sincerely looking forward to get support from the reviewer.

(5) in page 11, “It should be noted that the SEM images of cycled NCM-811 cathodes from both typical liquid electrolyte (Figure S14a and S14b) and quasi-solid electrolyte (Figure S14c and S14d) demonstrated similar morphologies as shown in Figure 3a and 3b, Figure 3e and 3f, respectively.” this conclusion is contradict to the results in the coin cell, where NCM cathode has CEI in liquid cell, while no CEI obtained in quasi-solid cell.

Thank you very much for your valuable question.

In fact, it should be Figure 4a and 4b, Figure 4e and 4f, not Figure 3a and 3b, Figure 3e and 3f as we have written in our original manuscript. We are terribly sorry for this careless mistake we have made in the original version.

Actually, SEM images shown in Figure S14 consisted well with that of the SEM images demonstrated in Figure 4. Even from our original Figure S14, apparent different cathode morphologies can be clearly observed already. To help the reviewer and potential readers to watch them more clearly, we have replaced two SEM images in our original Figure S14 with two new pictures with higher resolution (both were taken from the same electrodes) as shown in Figure R16. Clearly, for the cycled NCM-811 cathode harvested from the typical liquid electrolyte, apparent film-like CEI layer (yellow arrow marked in Figure R16a and 16b) was covered on the NCM-811 particle surface. It is worth noting that due to the irregular deposition of the by-products, some area also exhibited porous structure as highlighted by the yellow circles in Figure R16a. The original regular NCM-811 particles can hardly be observed. For sharp contrast, as shown in Figure R16 c and 16d, nearly no any CEI layer can be found on the NCM-811 cathode which was harvested from the quasi-solid electrolyte. Moreover, the original regular NCM-811 particles can be clearly observed. Therefore, the SEM images shown in Figure S14 consisted well with that of the SEM images demonstrated in Figure 4.

Figure R16 (now as Figure S14 in our revised Supplementary Information). SEM images of the cycled NCM-811 cathodes from NCM-811//Li pouch cells used (a, b) typical liquid electrolyte and (c, d) the prepared quasi-solid electrolyte that cycled under high temperature of 90 °C (yellow curves demonstrated in Figure 5c and Figure 5e).

We sincerely hope our explanation can eliminate the concerns and doubts from the reviewer on this question and looking forward to get support from the reviewer.

(6) in page 11, “The extraordinary pouch-cell performances harvested under such harsh condition (being damaged and tested under 90 oC high temperature) suggested the significant importance of using the prepared quasi-solid electrolyte in constructing highly-safe and efficient LMBs.” The conclusion should mention the reason leads to such significant importance.

Thank you very much for your valuable question and constructive suggestion.

We apologize for failing to provide a deeper explanation. And after kindly reminded by the reviewer, we summarized to main reasons for the significant importance of using the prepared quasi-solid electrolyte in constructing highly-safe and efficient LMBs.

In fact, this question is closely related to your 1st and 3rd comment. We think the excellent pouch-cell performances were also closely related to the properties of the quasi-solid electrolyte. As shown in our reply for your first comment, we think the unique behaviors/properties of the quasi-solid electrolyte reported in this work can be ascribed to the aggregative electrolyte configuration induced by the sub-nano confinement environment.

(1) The reason why pouch-cells assembled with quasi-solid electrolyte can work normally under 90 °C.

Benefiting from the sub-nano confinement environment constructed by the narrow MOF channels, tiny liquid confined inside the MOF channels (quasi-solid electrolyte) tend to possessed much higher decomposition temperature: The decomposition temperature (yellow highlighted) of liquid solvent within quasi-solid electrolyte encountered the biggest change: begun to decompose from nearly 200 °C, improved almost 100 °C compared with that of the typical liquid electrolyte. The salt decomposition temperature (light blue highlighted in both Figure 2e and Figure 2f) of quasi-solid electrolyte also experienced obvious increasement which started from nearly 400 °C, almost 50 °C higher than that of typical liquid electrolyte. The remarkably improved

decomposition temperature can be ascribed to the unique sub-nanoconfinement of the polar porous MOF host towards tiny amount of liquid electrolyte. This made the tiny amount of liquid electrolyte within the quasi-solid electrolytes hard to be evaporated. Due to the largely improved decomposition temperature of the quasi-solid electrolyte, pouch-cell assembled with quasi-solid electrolyte can work normally under 90 °C.

(2) The reason for good electrochemical performances of pouch-cells assembled with quasi-solid electrolyte under 90 °C.

We think the aggregative electrolyte configuration of the quasi-solid electrolyte contributed to the good electrochemical performances. Compared with typical diluent liquid electrolyte (Figure S3c and S3d), tiny liquid electrolyte confined inside MOF channels demonstrated more stronger Li-PC interaction and concentrated TFSI⁻ state, suggested more aggregative electrolyte configuration. It is worth noting that tiny liquid electrolyte confined inside MOF channels was even more aggregative than concentrated liquid electrolyte (3M LiTFSI-PC, Figure S3e and S3f). Due to the enhanced Li-PC solvents and Li-TFSI⁻ interactions, the solvation sheaths of lithium-ions became smaller but much compact. Correspondingly, the number of PC solvent molecules contained within each solvation sheath of the solvated lithium-ion (tiny liquid electrolyte confined inside MOF channels, contained only strongly coordinated PC solvents as shown in Figure 3a in our original manuscript) are greatly reduced compared to that of both the dilute electrolyte (which contained various free PC solvents and weakly-coordinated PC solvents within the solvated lithium-ions as exhibited in Figure S3c in our original Supplementary Information) and the concentrated electrolyte (contained both various weakly-coordinated and strongly-coordinated PC solvents within the solvation sheaths of solvated lithium-ions as exhibited in Figure S3e in our original Supplementary Information).

In this work, benefits from the eliminated free PC solvents and weakly-coordinated PC solvents as well as part of the strongly-coordinated PC solvents within solvated lithium-ions of the quasi-solid electrolyte, by-products related to the decomposition of PC solvent molecules are greatly reduced. Due to the greatly reduced side-reactions, the pouch-cells assembled with quasi-solid electrolyte demonstrated good cycling performances. The greatly improved decomposition temperature and the quasi-solid property of the flexible quasi-solid electrolyte made the pouch-cell work stably and safely even under harsh environment of high-temperature (90 °C) and being damages (bended and cut). And the corresponding discussion was also added and yellow highlighted in our revised manuscript.

“...Benefits from the eliminated free PC solvents and weakly-coordinated PC solvents as well as part of the strongly-coordinated PC solvents within solvated lithium-ions of the quasi-solid electrolyte (aggregative electrolyte configuration), by-products related to the decomposition of PC solvent molecules are greatly reduced. Due to the greatly reduced side-reactions, the pouch-cells assembled with quasi-solid electrolyte demonstrated good cycling performances. The greatly improved decomposition temperature and the quasi-solid property of the flexible quasi-solid electrolyte made the pouch-cell work stably and safely even under harsh environment of high-temperature (90 °C) and being damages (bended and cut)...” (page 12, in revised manuscript)

We sincerely appreciate your highly constructive suggestion and also sincerely looking forward to get support from the reviewer.

(7) Figure 1-3, too many texts and plots. The font size of number and texts in the plot are a bit small.

Thank you very much for your constructive suggestion.

After kindly suggested by the reviewer, we have streamlined the texts and some pictures within Figure 1-3 in our revised manuscript as much as possible. Yet, in order to make the Figures more accurate and easier for potential readers to understand, we hope to keep some necessary texts and did not revise them anymore. We sincerely hope you can understand this. Moreover, we have also modified the size of all the numbers and texts in our Figures and strive to keep the font size within all pictures consistent with each other.

Figure 1 (revised). The importance of using non-liquid electrolyte in constructing safe lithium-metal batteries. (a) Safety hazards which may induced by using traditional liquid electrolytes. (b) Radar chart of the advantages of quasi-solid electrolytes (compared with typical liquid electrolyte and solid-state electrolyte). (c) Lithium-metal battery assembled with proposed quasi-solid electrolyte.

In addition, due to the changes in Figure 1, we also revised the corresponding sentences in Introduction part and yellow highlighted in our newly revised manuscript.

“...Batteries, especially the lithium-ion batteries (LIBs)^{3, 4}, are widely used to power various electronic devices (from portable electric devices like hand watches to smart phones, laptops and even electric vehicles (EVs)). The rapid development and ever-increasing use of electronic equipment puts great demands on not only their densities and cycling lives, but the productions and shipments of LIBs. According to the report, both the global LIBs market and LIBs’s shipments have gradually increased during the past three years, and are also expected to steadily climbing in the coming five years⁵. Specifically, the global LIBs market and shipment in 2025 are predicted to be 108.9 billion dollars, 439.3 GWh, respectively⁵...” (page 3, in revised manuscript)

“...LMBs, especially the high-voltage LMBs which assembled with typical liquid electrolytes tend to suffer serious electrolyte degradations which were induced by the high reactivity between liquid electrolytes and charged transition metal oxide surfaces/reactive lithium-metals (Figure 1a)^{10, 20}...” (page 3, in revised manuscript)

“...As an intermediate state between liquid electrolyte and solid electrolyte, quasi-solid electrolyte would take the advantages of liquid electrolyte and solid electrolyte while avoiding the shortcomings of both sides (Figure 1b)^{24, 27}. Quasi-solid electrolyte, can not only provide mechanical stiffness to block dendrites, but deliver much safer operation environment (non-flammable) compared to typical liquid electrolyte. On the other hand, quasi-solid electrolyte also possesses higher ion conductivity and superior interfacial property than that of solid-state electrolyte. Confining lean amounts of liquid electrolyte inside the host matrix that possesses nano porous (sub-nano porous) structure as shown in Figure 1c is a promising method to prepare a quasi-solid electrolyte which can meet the above requirements discussed in Figure 1b (good contact towards electrodes, hard to be volatilized, stable and safe under high working temperature)...” (page 4, in revised manuscript)

Figure 2 (revised). Physical characterizations towards the MOF based quasi-solid electrolytes. (a) Schematic illustration of the precursor porous host material used for preparing under different conditions (from bottom to up: pristine MOF (CuBTC-PSS), activated MOF, MOF with liquid electrolyte coordinated inside its channels (defined as quasi-solid electrolyte)) and the corresponding X-ray powder diffraction (XRD) patterns of the three kinds of materials. (c) Pore size distributions of the activated MOF and MOF based quasi-solid electrolyte. (d) Thermogravimetric analysis (TGA) curve of the typical liquid electrolyte (1M LiTFSI-PC). (e) TGA curve of the MOF based quasi-solid electrolyte and (f) the enlarge version of (e). Schematic illustration of (g) the disadvantages of typical liquid electrolyte and (h) advantages of MOF based quasi-solid electrolyte.

Figure 3 (Revised). Physicochemical properties of the MOF based quasi-solid electrolyte. (a) Raman spectrum and (b) Fourier-transform infrared spectroscopy (FT-IR) of the prepared MOF based quasi-solid electrolyte. (c) Linear sweep voltammetry (LSV) curves of the two electrolytes (green curve: typical liquid electrolyte; blue curve: MOF based quasi-solid electrolyte). The inset schematically illustrates the different electrolyte configurations. (d) Comparison chart of thickness and quality of different electrolytes (quasi-solid electrolyte, commercial PP separator typical liquid electrolyte and commercial $\text{Li}_{1.3}\text{Al}_{0.3}\text{Ge}_{1.7}(\text{PO}_4)_3$ (LAGP) solid electrolyte). Photographs of ignition tests of (e) typical liquid electrolyte saturated glass fibers and the prepared MOF based quasi-solid electrolyte. (f) Ion conductivities and the corresponding activation energies (g) of the solid-state electrolyte, quasi-solid electrolyte and typical liquid electrolyte. (h) Advantages of the prepared quasi-solid electrolyte in constructing highly safe LMBs.

Figure 5 (revised). NCM-811//Li pouch-cells used quasi-solid electrolyte under harsh working condition (high temperature of 90 °C). (a) Schematic illustration of the cell structure and digital photo of NCM-811//Li pouch-cell assembled with MOF based quasi-solid electrolyte. (b) Discharge/charge curves and (c) the corresponding cycling performances of NCM-811//Li pouch-cell assembled with quasi-solid electrolyte under high cathode mass loading (about 20 mg cm⁻²) and high room temperature of 90 °C (yellow curve). The blue curve represents cycling performance obtained under room temperature of 25 °C. (d) Discharge/charge curves and (e) the corresponding cycling performances of NCM-811//Li pouch-cell assembled with typical liquid electrolyte under high cathode mass loading (about 20 mg cm⁻²) and room temperature of 90 °C (blue curve). The yellow curve represents cycling performance obtained under high temperature of 90 °C. (f) Cycling performances of NCM-811//Li pouch-cell (NCM-811 cathode mass loading about 20 mg cm⁻²) assembled with MOF based quasi-solid electrolyte under harsh working environment of high working temperature (90 °C) after being damaged (bended and cut).

We sincerely appreciate your highly constructive suggestion and also sincerely looking forward to get support from the reviewer.

REVIEWER COMMENTS

Reviewer #1 (Remarks to the Author):

The authors have addressed most the original questions, however, some important ones remain to be fixed:

1) The explanation for comment 6 (Reviewer 1) might be not reasonable. If no more liquid electrolyte was added into the cathodes, the electrolyte confined in the MOF pores seems to be impossible to go out from the MOF pores and then to wet the cathode surface, not to say to wet the inner part of the cathodes. IF the electrolyte in MOF can wet the cathode, it means the solvent molecules can go out, which will lose the feature of “nanoconfinement” through the cathode.

In addition, the mechanism of lean liquid electrolyte in typical LIBs is not suitable for the porous solid embedding electrolyte system. The MOFs confined electrolyte (solid) is different from the lean liquid electrolyte (liquid) because the former cannot flow like the common liquid electrolyte (if it is so tightly confined as described in this work). However, the latter can flow inside cathode pores and then wet the cathode particles although its volume is very low (e.g 3mL/Ah). On the other hand, if the MOFs trapped electrolytes can wet the cathodes, this indicates some electrolytes are not confined by the nanopores of porous solids, which should pose the same battery performance to the bulk/free electrolyte control as the surface and inner side of the electrode largely determine the battery performance. This is the most important fundamental question that determines the design principle.

Although the performance of the MOF confined system is proven to be much better than the bulk liquid electrolytes, clear understanding of why the cells work well without any liquid and solid-state electrolytes in the cathodes is necessary. Otherwise, the design will still present a magic.

Format issue:

Some format issues might need to be carefully corrected, e.g., it should not a page but an article number for the references of Nat. Commun.

Reviewer #2 (Remarks to the Author):

Authors have made great effort to address the questions raised from the revision. The revised manuscript has been improved significantly, hence I recommend to publish the manuscript.

Title: “A stable quasi-solid electrolyte boosting highly-efficient lithium-metal pouch-cell work safely under harsh environment”

Manuscript number: NCOMMS-21-31759A

Authors: Zhi Chang, Huijun Yang, Xingyu Zhu, Ping He and Haoshen Zhou*

We appreciate all these valuable comments from the reviewers. Following are our responses to these comments from two reviewers.

Response to Reviewers:

To Reviewer #1:2-16

To Reviewer #2:17

Reviewer #1 (Remarks to the Author):

The authors have addressed most the original questions, however, some important ones remain to be fixed:

Thank you for your positive comments on our efforts in responding to your comments and questions in the 1st round. We also thank you for providing the following suggestible comments. We also show our greatest respect to your precise scientific attitude, which really help us a lot to modify this current work. Thank you for your comments, suggestions, and scientific exchanges.

(1) The explanation for comment 6 (Reviewer 1) might be not reasonable. If no more liquid electrolyte was added into the cathodes, the electrolyte confined in the MOF pores seems to be impossible to go out from the MOF pores and then to wet the cathode surface, not to say to wet the inner part of the cathodes. IF the electrolyte in MOF can wet the cathode, it means the solvent molecules can go out, which will lose the feature of “nanoconfinement” through the cathode.

Thank you very much for your valuable questions and comments.

After carefully tasting your comments, we begun to realize that there were several essential misunderstandings, which severely influenced your judgement. We think the misunderstandings was of course due to our previous unclear narrative of our work. We are sorry about this and sincerely appreciate the opportunity that you have given to us to eliminate previous misunderstandings. We try our best to make it clear for the reviewer in the following discussion by supplementing additional important data and sincerely hope that subsequent revision can eliminate all the concerns and doubts from the reviewer on this question and satisfy the reviewer.

We think the first comment from the reviewer can be summarized in to two issues: (1) whether the electrolyte confined inside MOF can go out to wet the cathode; (2) if it can go out and wet the cathode, whether the “nanoconfinement” effect still exist.

Firstly, we think (1) parts of liquid electrolyte confined inside MOF can go out and wet the cathode. Secondly, we believe the (2) “nanoconfinement” effect always exist even after parts of liquid electrolyte go out during cycling. We have supplemented corresponding data and discussion to support our conclusion.

(1) For your **first question**: we think the concerns/doubts were caused by our inadequate and unclear description in both of our original and revised manuscript in the 1st round. We are sorry for not giving enough discussion in detail in our original manuscript. And we think the concerns/doubts from the reviewer can be answered after we explain the different electrolyte states that confined inside MOF channels.

Figure R1. XRD patterns of (a) pristine MOF, (b) activated MOF and (c) the prepared MOF-based quasi-solid electrolytes. (d) Schematically illustration of two types of liquid electrolytes inside the MOF channels. (e) BET curves of cycled MOF-based quasi-solid electrolytes after different cycles. (f) XRD patterns of pristine and cycled quasi-solid electrolytes.

We think parts of liquid electrolyte confined inside MOF can go out and wet the cathode. To figure out whether electrolyte inside MOF channels can go out or which part of the electrolyte came out from the MOF channel, we must firstly have a clear understanding of the states of the electrolyte inside the MOF channels.

As discussed in our previous response (as demonstrated in Figure R1 a-c), the (111) peak in the XRD patterns of MOF can be assigned to the liquid electrolyte-Cu metal interactions (suggesting the existence of **Type 2** electrolyte). Based on the data we have obtained shown in Figure R1e and R1f, and results reported by others, we think for the prepared MOF-based quasi-solid electrolyte, there are two types of liquid electrolytes inside the MOF channels: **Type 1 electrolyte**: the physically confined liquid electrolyte inside MOF channels (solely confined inside the sub-nano channels of MOF) and **Type 2 electrolyte**: simultaneous physically & chemically confined liquid electrolyte (chemically coordinated by unsaturated Cu metal sites inside MOF channels while also physically confined inside the sub-nano channels). It worth noting that as schematically demonstrated in Figure R1d, both **Type 1** and **Type 2** electrolyte were under the sub-nanoconfinement effect constructed by the narrow MOF channels.

Then, we think only type 1 electrolyte can go out and wet the cathode while the type 2 electrolyte did not go out the MOF channels.

Considering the additional chemically coordination between the **Type 2** electrolyte and the Cu metal sides inside MOF channels, we think **Type 2** electrolyte was less likely to go out from the MOF channels when compared with **Type 1** electrolyte, which was only physically confined inside MOF channels. The changing BET curves of cycled MOF-based quasi-solid electrolytes after different cycles in Figure R1f (highlighted by the red arrow) suggested different amount of liquid electrolyte was constantly consumed during cycling. We think the constantly consumed

electrolyte comes from the **Type 1** electrolyte which was only under physical confinement. The difference (highlighted by the blue arrow) between the BET of activated MOF (blue curves in Figure R1f) and the BET of the cycled MOF-based quasi-solid electrolyte (the dark green curves) can be ascribed to the existence of the remained electrolyte. In addition, as clearly shown in Figure R1e, during the whole charge/discharge process, the **Type 2** electrolyte can always be detected, which further indicated the **Type 2** electrolyte did not go out the MOF channels.

Therefore, we think only **Type 1** electrolyte can go out the MOF channels and then wet the cathode as schematically illustrated in Figure R2.

Figure R2. Schematic illustration of the prepared MOF-based quasi-solid electrolyte in wetting the NCM-811 cathode.

Figure R3. Tap peeling method used towards cycled NCM-811 cathode (after 700 cycles) to verify the NCM-811 cathode was wetted by electrolyte from the MOF channels. (a) Schematically illustration of the Tap peeling method employed to peel off certain amounts of NCM-811 materials and expose the internal area of cycled NCM-811 cathode for the following Raman experiment. SEM images of the cycled NCM-811 cathode under Tap peeling test for (b) 0 time, (c) 1 time, (d) 3 times, (e) 5 times, (f) 7 times and (g) 9 times.

To verify the NCM-811 cathode can be wetted by liquid electrolyte came out from the MOF channels, we have supplemented important data by detecting the Raman spectra of cycled NCM-811 cathode under different depths after different times of Tap peeling test as schematically illustrated in Figure R3a. By constantly peeling off the surface layer of cycled NCM-811 cathode, the internal sections in deep depths of NCM-811 can be exposed. Then, Raman measurement was

applied to detect whether there were liquid electrolyte peaks contained inside the deep depth of the cycled NCM-811 cathode. As shown in the SEM images of Figure R3, NCM-811 cathodes with different thicknesses were successfully obtained after different times of Tap peeling experiment. The corresponding Raman spectrum were then tested and demonstrated in Figure R4a.

Figure R4. (a) Raman spectra detected from cycled NCM-811 cathode under different times of Tap peeling test (from 0 to 9 times). (b) Raman spectrum collected from the newly prepared MOF-based quasi-solid electrolyte. (c) The Raman color mapping of (a).

Obviously, during the whole Tap peeling process, there were two apparent peaks can be detected (Figure R4a, highlighted by the red dotted rectangles), which can be ascribed to the electrolyte related peaks as shown in Figure R4b (redraw from Figure 3a in our original manuscript). The corresponding color mapping picture in Figure R4c also indicated the same result, although the intensities of the two peaks decreased gradually. These results together suggested that the liquid electrolyte came out from the MOF channels can wet the cathode even into the deep depth.

In short summary, based on the data that we have supplemented, parts of liquid electrolyte (only physically confined inside the MOF channels) confined inside MOF can go out and wet the cathode.

(2) For your second question: we think the “nanoconfinement” effect always exist even after parts of liquid electrolyte go out during cycling.

As we have discussed in Figure R1 (also in Figure R5a), we think for the prepared MOF-based quasi-solid electrolyte, there are two types of liquid electrolytes inside the MOF channels: **Type 1 electrolyte:** the physically confined liquid electrolyte inside MOF channels (solely confined inside the sub-nano channels of MOF) and **Type 2 electrolyte:** simultaneous physically & chemically confined liquid electrolyte (chemically coordinated by unsaturated Cu metal sites inside MOF channels while also physically confined inside the sub-nano channels).

Therefore, as schematically demonstrated in Figure R1d, both **Type 1** and **Type 2** electrolyte were under the sub-nanoconfinement effect constructed by the narrow MOF channels.

In our reply to your first question, we have firmly verified that only Type 1 electrolyte can go out and wet the cathode while the Type 2 electrolyte did not go out the MOF channels (as schematically illustrated in Figure R5b).

The mechanism of how the “sub-nano confinement” effect maintained during the electrochemical cycling processes was also proposed and shown in Figure R5c.

As demonstrated in Figure R5c, during cycling, especially at the initial stage, only very small amount of **Type 1** electrolyte went out from the MOF channels to wet the NCM-811 cathode. As the electrochemical process continues, more **Type 1** electrolyte go out the MOF channels This dynamic process will stop until all the **Type 1** electrolyte that originally inside the MOF channels fully go out and be totally consumed. Therefore, during this dynamic process, there are remaining **Type 1** and the unreduced/unchanged amount **Type 2** electrolyte inside the MOF channels before all the **Type 1** electrolyte is totally consumed. Thus, the sub-nanoconfinement will always exist during the multiple electrochemical cycling processes. More importantly, even if the **Type 1** electrolyte is totally consumed, while the **Type 2** electrolyte did not go out from the MOF channels (as schematically illustrated in Figure R5b), the sub-nano confinement effect will always exist. Therefore, we think whether the liquid electrolyte go out the MOF channels or not, it is not contradicts with the existence of the sub-nanoconfinement effect.

Figure R5. (a) Schematic illustration of two types of liquid electrolytes inside the MOF channels. (b) Schematic illustration of working mechanism for the MOF-based quasi-solid electrolyte in this work. (c) How the “sub-nano confinement” effect maintained during the electrochemical cycling processes.

This conclusion can also be perfectly supported by the BET and XRD data shown in Figure R1 e and R1f. To make it clearer, we further reorganized the Figure R5 and Figure R1 into Figure R6. Clearly, as shown in Figure R6b, the gap between activated MOF and the newly prepared MOF-based quasi-solid electrolyte represented the overall amount of electrolyte (both Type 1 and Type 2 electrolyte) confined inside MOF channels. After cycled for 100, 300 and 500 cycles,

electrolyte inside MOF channels decreased gradually, which can be ascribed to the constantly consuming of Type 1 electrolyte. The gradually consumed electrolyte also further indicated during cycling, there were remaining Type 1 (and the unreduced/unchanged amount Type 2 electrolyte) inside the MOF channels before all the Type 1 electrolyte is totally consumed. For example, after cycled for 100 cycles, small amount of Type 1 electrolyte was consumed. However, there was still enough Type 1 electrolyte remained for the following 300, 500, and 700 cycles. The same conclusion can be made when cycled for 300 and 500 cycles. The gap between activated MOF and the MOF-based quasi-solid electrolyte after 700 cycles represented the remained electrolyte inside the MOF channels. Thus, the sub-nanoconfinement (only for Type 1 electrolyte) will always exist during the multiple electrochemical cycling processes. More importantly, even if the Type 1 electrolyte is totally consumed, while the Type 2 electrolyte did not go out from the MOF channels (indicated by the (111) peaks of different MOF-based quasi-solid electrolyte after cycling), the sub-nanoconfinement effect will always exist.

Therefore, we think whether the liquid electrolyte go out the MOF channels or not, it is not contradicts with the existence of the sub-nanoconfinement effect.

Figure R6. (a) Schematic illustration of working mechanism for the MOF-based quasi-solid electrolyte in this work, and their corresponding (b) BET curves and (c) XRD patterns.

In summary, after supplemented those data, it is reasonable for us to conclude that parts of liquid electrolyte confined inside MOF can go out and wet the cathode while the “nanoconfinement” effect always exist even after parts of liquid electrolyte go out during cycling.

In addition, since those added field data are important in helping us explaining the core design principle of this work, we decide to supplement the corresponding results into our newly revised manuscript and supplementary information. As showing below, Figure R7 was added as Figure 5 into our newly revised manuscript. And Figure R5 and Figure R8 were supplemented as Figure S13 and Figure S14 into our revised supplementary information, respectively. We also added the corresponding discussion, the related experiment section and highlighted them in our newly revised manuscript and Supplementary Information.

Figure R7 (Now as Figure 5 in our newly revised manuscript). Tape peeling method used towards cycled NCM-811 cathode (after 700 cycles) to verify the NCM-811 cathode was wetted by electrolyte from the MOF channels. (a) Schematically illustration of the Tape peeling method employed to peel off certain amounts of NCM-811 materials and expose the internal area of cycled NCM-811 cathode for the following Raman experiment. SEM images of the cycled NCM-811 cathode under Tape peeling test for (b) 0 time, (c) 1 time, (d) 3 times, (e) 5 times, (f) 7 times and (g) 9 times. (h) Raman spectra detected from cycled NCM-811 cathode under different times of Tape peeling test (from 0 to 9 times) and the corresponding (i) color mapping. (j) XRD patterns of pristine and cycled quasi-solid electrolytes.

“...Noting that those characterizations were only focused on the surface of NCM-811 cathode. To further understand the working mechanism of using the prepared MOF-based quasi-solid electrolyte in lithium-metal batteries, detail information inside the deep depths of the cycled cathode needs to be further studied. Therefore, cycled NCM-811 cathode collected from the NCM-811/Li used quasi-solid electrolyte (after 700 cycles, harvested from the cell demonstrated in Figure S11) was studied by Raman. As schematically demonstrated in Figure 5a, to collect detail and accurate information even inside the deep depths of the cycled NCM-811 cathode, a special tape peeling test was used to peel off the surface layers of the cycled NCM-811 cathode and thus exposed the new NCM-811 cathode interphases to the Raman laser. After the different times of tap peeling (from 0 to 9 times), new interphases of NCM-811 cathode under different depths (different thicknesses, Figure 5b-5g) were hence obtained. Then the obtained Raman spectra from each depth were under further investigation. Obviously, as demonstrated in Figure 5h, two apparent peaks which were related to the liquid electrolyte, can be constantly detected under all the depths. Moreover, the two peaks maintained almost the same shapes as the shapes of liquid electrolyte confined inside the MOF channels of the quasi-solid electrolyte. This suggested that liquid electrolyte can go out from the MOF channels of the quasi-solid electrolyte and consequently wet the NCM-811 cathode into the deep depth despite the intensities of the

electrolyte related peaks decreased gradually (Figure 5i). In addition, as schematically illustrated in Figure S13a, there were two types of liquid electrolytes confined inside the MOF channels: one is the liquid electrolyte under only physical confinement (type 1), the other is the liquid electrolyte under both physical and chemical confinement (type 2). We thought the physically confined liquid electrolyte (type 1) went out from the MOF channels and wetted the cathode (as schematically demonstrated in Figure S13b and Figure S14a). The gradually increased BET results shown in Figure S14b indicated the constantly consuming of liquid electrolyte. Since there were apparent (111) peaks (indicated the chemically bonded liquid electrolyte) can still be found in the XRD patterns of the cycled MOF-based quasi-solid electrolytes (Figure 5j), which suggested liquid electrolyte under both physical and chemical confinement (type 2) cannot go out from the MOF channels to wet the cathode. Therefore, the constantly consumed liquid electrolyte can only be ascribed to the physically confined liquid electrolyte (type 1) inside the MOF channels. Also, as schematically illustrated in Figure S13c, since there still existed both physically and chemically confined liquid electrolyte (type 2) inside the MOF channels, the sub-nanoconfinement was constantly existed during the whole electrochemical cycling process...” (page 11, revised Manuscript)

The specific surface area (SSA) the cycled MOF-based quasi-solid electrolytes after different cycles was determined based on Brunauer-Emmett-Teller (BET) theory in the relative pressure range of 0.04 to 0.2. (page 6, revised Supplementary Information)

Fourier-transform infrared (FTIR) Characterization for cycled NCM-811 cathode

To verify the NCM-811 cathode can be wetted by liquid electrolyte came out from the MOF channels, the Raman spectra of cycled NCM-811 cathode under different depths after different times of Tap peeling test were collected. By constantly peeling off the surface layer of cycled NCM-811 cathode, the internal sections in deep depths of NCM-811 can be exposed. Then, Raman measurement was applied to detect whether there were liquid electrolyte peaks contained inside the deep depth of the cycled NCM-811 cathode. (page 7, revised Supplementary Information)

Figure R5 (Now as Figure S13 in our revised Supplementary Information). (a) Schematic illustration of two types of liquid electrolytes inside the MOF channels. (b) Schematic illustration of working mechanism for the MOF-based quasi-solid electrolyte in this work. (c) How the “sub-nano confinement” effect maintained during the electrochemical cycling processes.

“...As shown in Figure S13, it was thought for the prepared MOF-based quasi-solid electrolyte, there are two types of liquid electrolytes inside the MOF channels: Type 1 electrolyte: the physically confined liquid electrolyte inside MOF channels (solely confined inside the sub-nano channels of MOF) and Type 2 electrolyte: simultaneous physically & chemically confined liquid electrolyte (chemically coordinated by unsaturated Cu metal sites inside MOF channels while also physically confined inside the sub-nano channels). Therefore, as schematically demonstrated in Figure S13a, both Type 1 and Type 2 electrolyte were under the sub-nanoconfinement effect constructed by the narrow MOF channels. It was thought that only Type 1 electrolyte can go out and wet the cathode while the Type 2 electrolyte did not go out the MOF channels (as schematically illustrated in Figure S13b). The mechanism of how the “sub-nano confinement” effect maintained during the electrochemical cycling processes was also proposed and shown in Figure S13c. As demonstrated in Figure S13c, during cycling, especially at the initial stage, only very small amount of Type 1 electrolyte went out from the MOF channels to wet the NCM-811 cathode. As the electrochemical process continues, more Type 1 electrolyte go out the MOF channels This dynamic process will stop until all the Type 1 electrolyte that originally inside the MOF channels fully go out and be totally consumed. Therefore, during this dynamic process, there are remaining Type 1 and the unreduced/unchanged amount Type 2 electrolyte inside the MOF channels before all the Type 1 electrolyte is totally consumed. Thus, the sub-nanoconfinement will always exist during the multiple electrochemical cycling processes. More importantly, even if the Type 1 electrolyte is totally consumed, while the Type 2 electrolyte did not go out from the MOF channels (as schematically illustrated in Figure S13b), the sub-nano confinement effect will always exist...” (page 21, revised Supplementary Information)

Figure R8 (Now as Figure S14 in our revised Supplementary Information). (a) Schematic illustration of working mechanism for the MOF-based quasi-solid electrolyte in this work. (b) BET curves of cycled MOF-based quasi-solid electrolytes after different cycles.

“...As shown in Figure S14a (similar as Figure S13b), only Type 1 electrolyte can go out and wet the cathode while the Type 2 electrolyte did not go out the MOF channels. Clearly, as shown in Figure S14b, the gap between activated MOF and the newly prepared MOF-based quasi-solid electrolyte represented the overall amount of electrolyte (both Type 1 and Type 2 electrolyte) confined inside MOF channels. After cycled for 100, 300 and 500 cycles, electrolyte inside MOF channels decreased gradually, which can be ascribed to the constantly consuming of Type 1 electrolyte. The gradually consumed electrolyte also further indicated during cycling, there were remaining Type 1 (and the unreduced/unchanged amount Type 2 electrolyte) inside the MOF channels before all the Type 1 electrolyte is totally consumed. For example, after cycled for 100 cycles, small amount of Type 1 electrolyte was consumed. However, there was still enough Type 1 electrolyte remained for the following 300, 500, and 700 cycles. The same conclusion can be made when cycled for 300 and 500 cycles. The gap between activated MOF and the MOF-based quasi-solid electrolyte after 700 cycles represented the remained electrolyte inside the MOF channels. Thus, the sub-nanoconfinement (only for Type 1 electrolyte) will always exist during the multiple electrochemical cycling processes. More importantly, even if the Type 1 electrolyte is totally consumed, while the Type 2 electrolyte did not go out from the MOF channels (indicated by the (111) peaks of different MOF-based quasi-solid electrolyte after cycling shown in Figure 5i), the sub-nano confinement effect will always exist...” (page 21, revised Supplementary Information)

We sincerely appreciate the constructive suggestion from the reviewer for helping our work more logical and complete. We also sincerely hope our explanation can satisfied the reviewer on this question and are sincerely looking forward to get support from the reviewer.

(2) In addition, the mechanism of lean liquid electrolyte in typical LIBs is not suitable for the porous solid embedding electrolyte system. The MOFs confined electrolyte (solid) is different from the lean liquid electrolyte (liquid) because the former cannot flow like the common liquid electrolyte (if it is so tightly confined as described in this work). However, the latter can flow inside cathode pores and then wet the cathode particles although its volume is very low (e.g 3mL/Ah). On the other hand, if the MOFs trapped electrolytes can wet the cathodes, this indicates some electrolytes are not confined by the nanopores of porous solids, which should pose the same battery performance to the bulk/free electrolyte control as the surface and inner side of the electrode largely determine the battery performance. This is the most important fundamental question that determines the design principle.

Although the performance of the MOF confined system is proven to be much better than the bulk liquid electrolytes, clear understanding of why the cells work well without any liquid and solid-state electrolytes in the cathodes is necessary. Otherwise, the design will still present a magic.

Thank you very much for your valuable questions and comments.

Actually, this comment was closely related to your first comment. This comment can also be summarized into two questions: (1) whether the electrolyte confined inside MOF can go out to wet the cathode; (2) if it can go out and wet the cathode, whether this part of electrolyte which came out from the MOF channels possess the same electrolyte configuration (consequently same electrochemical performances) as that of conventional bulk/free electrolyte.

(1) The first question is actually **same as the 1st comment** from the reviewer. We have explained in detail in our response to the 1st comment from the reviewer. As we have discussed in detail in our reply to your first comment, by adding several important data (from Figure R1 to Figure R8), it was confirmed that parts of liquid electrolyte confined inside MOF can go out (flow to the depth of cathode) and wet the cathode while the “nanoconfinement” effect always exist even after parts of liquid electrolyte go out during cycling. Therefore, the reviewer is suggested to find our detail explanation in our response to the 1st comment from the reviewer.

(2) For your second question: we think part of electrolyte that came out from the MOF channels possessed much aggregative electrolyte configuration than conventional bulk/free electrolyte. As demonstrated in the Cell Assembly and Electrochemical Measurements section in our Supplementary Information, the prepared quasi-solid electrolytes were closely attached to the cathodes and followed by a physical pressing process. Therefore, close contact between the NCM-811 cathode and MOF-based quasi-solid electrolyte was achieved as shown in Figure R9. Benefits from the close contact between the NCM-811 cathode and MOF-based quasi-solid electrolyte, part of electrolyte confined inside MOF channels (Type 1 electrolyte) directly contacted with the NCM-811 cathode and then wetted the cathode without changing its original aggregative configuration (formed inside MOF channels as demonstrated in Figure 3a and 3b in our original manuscript). As exhibited in Figure R10, there were apparent liquid electrolyte related Raman peaks can be detected inside the different depths of cycled NCM-811 cathode. Obviously, compared with typical diluent electrolyte and concentrated electrolyte (Figure R10a and b, redraw from Figure S3 in our original Supplementary Information), electrolyte detected inside the depths of cycled NCM-811 cathode demonstrated similar peaks as that of electrolyte confined inside the MOF channels (as shown in Figure R10 c and d). Thus, electrolyte went out from the MOF channels possessed aggregative than that of conventional bulk/free electrolyte. Due to its unique electrolyte configuration (aggregative electrolyte configuration) and special physicochemical properties, we have obtained CEI-free cathode (as shown in Figure 4e-h in our original Manuscript) and dendritic-free lithium anode (as shown in Figure S12 in our original Supplementary Information). This suggested that the liquid electrolyte consumed by the side-reactions occurred on the electrode/electrolyte interphases were remarkably suppressed. Therefore, excellent electrochemical performances were finally obtained.

We sincerely hope our explanation can satisfied the reviewer on this question this time and are sincerely looking forward to get support from the reviewer.

Figure R9. Schematic illustration of the close contact between the prepared MOF-based quasi-solid electrolyte and the NCM cathode and the following cathode wetting process.

Figure R10. Raman spectra of (a) 1M typical liquid, (b) concentrated electrolyte and (c) of the prepared MOF based quasi-solid electrolyte. (d) Raman spectra detected from cycled NCM-811 cathode under different times of Tap peeling test (from 0 to 9 times).

Format issue:

Some format issues might need to be carefully corrected, e.g., it should not a page but an article number for the references of Nat. Commun.

Thank you very much for your valuable suggestion. We are sorry for this careless mistake. After kindly suggested by the reviewer, we have revised the format of all the references and yellow highlighted in our newly revised version.

References

1. Armand, M. & Tarascon, J.M. Building better batteries. *Nature* **451**, 652-657 (2008).
2. Dunn, B., Kamath, H. & Tarascon, J.M. Electrical Energy Storage for the Grid: A Battery of Choices. *Science* **334**, 928-935 (2011).

3. Tarascon, J. & Armand, M. Issues and challenges facing rechargeable lithium batteries. *Nature* **414**, 359-367 (2001).
4. Goodenough, J.B. & Kim, Y. Challenges for Rechargeable Li Batteries. *Chem. Mater.* **22**, 587-603 (2010).
5. A. Gupta & N. Paranjape. Lithium Ion Battery Market Outlook 2020-2026. Share Analysis. Report ID: GMI1135, 510 page, 2020. https://www.gminsights.com/industry-analysis/lithium-ion-battery-market?utm_source=prnewswire.com&utm_medium=referral&utm_campaign=Paid_prnewswire.
6. Chen, S. et al. High-Efficiency Lithium Metal Batteries with Fire-Retardant Electrolytes. *Joule* **2**, 1548-1558 (2018).
7. Jiao, S. et al. Stable cycling of high-voltage lithium metal batteries in ether electrolytes. *Nat. Energy* **3**, 739-746 (2018).
8. Liu, J. et al. Pathways for practical high-energy long-cycling lithium metal batteries. *Nat. Energy* **4**, 180-186 (2019).
9. Niu, C. et al. High-energy lithium metal pouch cells with limited anode swelling and long stable cycles. *Nat. Energy* **4**, 551-559 (2019).
10. Qian, J. et al. High rate and stable cycling of lithium metal anode. *Nat. Commun.* **6**, 6362 (2015).
11. Zeng, Z. et al. Non-flammable electrolytes with high salt-to-solvent ratios for Li-ion and Li-metal batteries. *Nat. Energy* **3**, 674-681 (2018).
12. Zheng, J. et al. Electrolyte additive enabled fast charging and stable cycling lithium metal batteries. *Nat. Energy* **2**, 17012 (2017).
13. Fan, X. et al. Non-flammable electrolyte enables Li-metal batteries with aggressive cathode chemistries. *Nat. Nanotechnol.* **13**, 715-722 (2018).
14. Suo, L. et al. Fluorine-donating electrolytes enable highly reversible 5-V-class Li metal batteries. *Proc. Natl. Acad. Sci. U S A.* **115**, 1156-1161 (2018).
15. Wang, J. et al. Superconcentrated electrolytes for a high-voltage lithium-ion battery. *Nat. Commun.* **7**, 12032 (2016).
16. Yamada, Y. et al. Hydrate-melt electrolytes for high-energy-density aqueous batteries. *Nat. Energy* **1**, 1-9 (2016).
17. Zheng, Q. et al. A cyclic phosphate-based battery electrolyte for high voltage and safe operation. *Nat. Energy* **5**, 291-298 (2020).
18. Xue, W. et al. FSI-inspired solvent and “full fluorosulfonyl” electrolyte for 4 V class lithium-metal batteries. *Energy Environ. Sci.* **13**, 212-220 (2020).
19. Li, M., Wang, C., Chen, Z., Xu, K. & Lu, J. New concepts in electrolytes. *Chem. Rev.* **120**, 6783-6819 (2020).
20. Zhang, X.Q. et al. A Sustainable Solid Electrolyte Interphase for High-Energy-Density Lithium Metal Batteries Under Practical Conditions. *Angew. Chem. Inter. Ed.* **132**, 3278-3283 (2020).
21. Chang, Z. et al. A stable high-voltage lithium-ion battery realized by an in-built water scavenger. *Energy Environ. Sci.* **13**, 1197-1204 (2020).
22. Chang, Z. et al. Beyond the concentrated electrolyte: further depleting solvent molecules within a Li⁺ solvation sheath to stabilize high-energy-density lithium metal batteries. *Energy Environ. Sci.* **13**, 4122-4131 (2020).

23. Chen, T. et al. Ionic liquid-immobilized polymer gel electrolyte with self-healing capability, high ionic conductivity and heat resistance for dendrite-free lithium metal batteries. *Nano Energy* **54**, 17-25 (2018).
24. Choudhury, S., Mangal, R., Agrawal, A. & Archer, L.A. A highly reversible room-temperature lithium metal battery based on crosslinked hairy nanoparticles. *Nat. Commun.* **6**, 10101 (2015).
25. Duan, H. et al. Dendrite-Free Li-Metal Battery Enabled by a Thin Asymmetric Solid Electrolyte with Engineered Layers. *J. Am. Chem. Soc.* **140**, 82-85 (2018).
26. Eshetu, G.G. et al. Ultrahigh Performance All Solid-State Lithium Sulfur Batteries: Salt Anion's Chemistry-Induced Anomalous Synergistic Effect. *J. Am. Chem. Soc.* **140**, 9921-9933 (2018).
27. Lu, Y., Tu, Z. & Archer, L.A. Stable lithium electrodeposition in liquid and nanoporous solid electrolytes. *Nat. Mater.* **13**, 961-969 (2014).
28. Chi, X. et al. A highly stable and flexible zeolite electrolyte solid-state Li–air battery. *Nature* **592**, 551-557 (2021).
29. Ye, L. & Li, X. A dynamic stability design strategy for lithium metal solid state batteries. *Nature* **593**, 218-222 (2021).
30. Chang, Z. et al. Fabricating better metal-organic frameworks separators for Li–S batteries: Pore sizes effects inspired channel modification strategy. *Energy Storage Mater.* **25**, 164-171 (2020).
31. Shen, L. et al. Creating Lithium- Ion Electrolytes with Biomimetic Ionic Channels in Metal–Organic Frameworks. *Adv. Mater.* **30**, 1707476 (2018).
32. Otake, K.-i. et al. Confined water-mediated high proton conduction in hydrophobic channel of a synthetic nanotube. *Nat. Commun.* **11**, 843 (2020).
33. Rieth, A. J., Hunter, K. M., Dincă, M. & Paesani, F. Hydrogen bonding structure of confined water templated by a metal-organic framework with open metal sites. *Nat. Commun.* **10**, 4771 (2019).
34. Ichii, T. et al. Observation of an exotic state of water in the hydrophilic nanospace of porous coordination polymers. *Commun. Chem.* **3**, 1–6 (2020).
35. Cai, G. et al. Sub-nanometer confinement enables facile condensation of gas electrolyte for low-temperature batteries. *Nat. Commun.* **12**, 3395 (2021).
36. Xu, K. Nonaqueous liquid electrolytes for lithium-based rechargeable batteries. *Chem. Rev.* **104**, 4303-4418 (2004).
37. Chen, S. et al. High-efficiency lithium metal batteries with fire-retardant electrolytes. *Joule* **2**, 1548-1558 (2018).
38. Cho, S.-J. et al. Nonflammable Lithium Metal Full Cells with Ultra-high Energy Density Based on Coordinated Carbonate Electrolytes. *iScience* **23**, 100844 (2020).
39. Fan, X. et al. Non-flammable electrolyte enables Li-metal batteries with aggressive cathode chemistries. *Nat. Nanotechnol.* **13**, 715-722 (2018).
40. Chang, Z. et al. A Liquid Electrolyte with De-Solvated Lithium Ions for Lithium-Metal Battery. *Joule* **4**, 1776-1789 (2020).

Special reply to reviewer 1:

Finally, we sincerely appreciate the reviewer for his/her rigorous attitude towards scientific research. We also highly appreciate your constructive and valuable advices and suggestions. Inspired by your valuable comments and suggestions, we have tried our best to find reasonable and persuasive evidences to express our point of view more clearly and at the same time push our article into a deeper level. Frankly speaking, we did not think deeply on these issues before the reviewers asked us. More importantly, not only limited to this work, your insightful questions and suggestions can also help us solving other thorny but important problems, which were originally difficult to explain in several our ongoing works. Therefore, we sincerely appreciate your highly constructive questions and suggestions as well as your rigorous attitude. Thank you very much, they help us a lot. We sincerely hope that you will be satisfied with our sincere explanation and corresponding amendments. Thank you for all your comments and suggestions during the previous and current rounds of peer review.

Reviewer #2 (Remarks to the Author):

Authors have made great effort to address the questions raised from the revision. The revised manuscript has been improved significantly, hence I recommend to publish the manuscript.

We are appreciating to know you are satisfied with our revision. We also thank you very much for your positive comment and your recommendation on this article. Thank you for your comments, suggestions, and scientific exchanges.

REVIEWERS' COMMENTS

Reviewer #1 (Remarks to the Author):

This revision has reasonably addressed the inconsistency on the "nanoconfinement" effect, and this reviewer would support the publication of the current version.

Title: “A stable quasi-solid electrolyte boosting highly-efficient lithium-metal pouch-cell work safely under harsh environment”

Manuscript number: NCOMMS-21-31759B

Authors: Zhi Chang, Huijun Yang, Xingyu Zhu, Ping He and Haoshen Zhou*

We appreciate all these valuable comments from the reviewers. Following are our responses to these comments from two reviewers.

Response to Reviewers:

To Reviewer #1:2

Reviewer #1 (Remarks to the Author):

This revision has reasonably addressed the inconsistency on the "nanoconfinement" effect, and this reviewer would support the publication of the current version.

We are appreciating to know you are satisfied with our revision. We also thank you very much for your positive comment and your recommendation on this article. Thank you for your comments, suggestions, and scientific exchanges.